# The Sky Is The Limit When Clustering Is Equated With Disentanglement

## Abstract

Disentangled representation learning allows data to be mapped to a latent space where factors of variation can be individually manipulated. These factors define a direct notion of similarity between observations that naturally groups them into clusters with shared factors of variation. While this has been empirically shown to be effective on simple datasets, it is unclear how or when complex real-world data can be disentangled into representations that allow the same degree of manipulation and clustering. To advance the field of disentangled representation learning and clustering, we provide a new theoretical perspective by equating disentanglement with clustering by using factors of variation as a measure of element-wise similarity. This leads to a simple yet important observation: Instead of explicitly clustering the elements of a dataset, we can implicitly cluster them by learning to represent and generate the elements of each cluster. Furthermore, this observation reveals that implicit clusters have a lower bound because (I) explicit clusters are a subset of implicit clusters, and (II) implicit clusters can generate novel elements not present in the finite dataset through combinatorial generalization. Building on these insights, we derive an implicit neural clustering approach based on identifying factors of variation in the latent space. We validate our findings through experiments on synthetic image data and empirical evidence from related state-of-the-art works. This demonstrates the practical relevance of our approach and promising potential for synthesizing complete datasets from limited data, addressing data distribution gaps, improving interpretability in cluster analysis, enhancing SSL and classification tasks, and reducing data storage space.

## 1 Introduction

Understanding and controlling the underlying factors of variation in data is central to disentangled representation learning (Wang et al., 2022). Disentangled latent spaces not only group similar elements naturally into clusters (Ding et al., 2022) but also allow precise data manipulation when combined with a generator (Higgins et al., 2017). Recent advances in deep generative clustering have shown the potential both to learn disentangled representations and cluster data apoints simultaneously, enabling the generation of high-quality synthetic data (Chen et al., 2016; Mukherjee et al., 2019; Lee et al., 2020; Yu & Welch, 2021; Ding et al., 2022). These approaches move away from traditional clustering algorithms, which rely on explicit partitioning based on learned or handcrafted features, towards generative models that leverage disentangled representations. In this context, generative models for controllable image synthesis, such as GANs (Karras et al., 2020; Brock et al., 2019) and Diffusion models (Rombach et al., 2022; Croitoru et al., 2023), now produce synthetic images realistic enough to improve downstream tasks like classification (Azizi et al., 2023; Fan et al., 2023) and can help self-supervised learning (SSL) methods learn better general purpose embeddings (Chai et al., 2021; Jahanian et al., 2021; Tian et al., 2024). While prior work on deep generative clustering has focused primarily on improving clustering in the traditional sense, a more implicit approach has the potential to synthesize full datasets and in turn fill gaps in data distributions, improve cluster interpretability, reduce storage needs, or enhance SSL embeddings and classification tasks. This leads to our main research question: *What if instead of explicitly clustering the elements of a dataset, we could represent and generate these clusters implicitly?*

To answer this question, we put forward a simple sampling method from a disentangled latent space, which we call *Implicit Neural Clustering*. Rather than explicitly assigning data points to clusters, we implicitly represent and generate the clusters with controllable factors of variation. With *Implicit Neural Clustering*, we can theoretically equate clustering and generative models with disentanglement, which leads to an *implicit neural* perspective of clustering. Previous work by Zhao et al. (2020); Ding et al. (2022) has already pointed out that using factors of variation as a measure of similarity will naturally group data into clusters. We move a step further and stress that clusters emerge naturally from controlling factors of variation in the latent space, i.e., control over a disentangled semantic latent space inherently dictates cluster member-

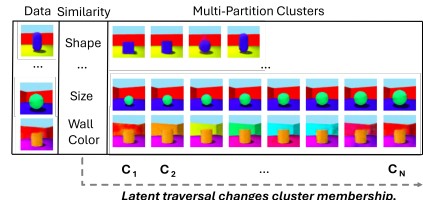

Figure 1: Latent traversals of factors of variation changes cluster memberships.

ship (see Figure 1). This shift in perspective provides two key theoretical insights: (I) explicit clusters are a subset of implicit clusters, and (II) implicit clusters have a *lower bound*, which is given by the ability of an encoder to disentangle factors of variation and by the realism as well as combinatorial generalization of a controllable generator.

*Implicit Neural Clustering* samples and modifies encoded disentangled representations of elements in a dataset with atomic group actions to implicitly generate clusters. On top of the group-based definition of disentangled representations by Higgins et al. (2018), we define an atomic group action as a partition of a latent traversal direction. After an atomic group action modifies a latent representation, a generator produces a data point reflecting this change while keeping other factors unchanged. Using disentangling variational autoencoder (VAEs), we show that atomic group actions exist and can be identified using Kernel Density Estimation (KDE) as a partitioning algorithm on each dimension of the disentangled representations of dataset elements separately. Furthermore, probing for atomic group actions leads us to an effective and simple qualitative measure for disentanglement, which is more informative than the commonly-used disentanglement visualization with Hinton Matrices (Eastwood & Williams, 2018; Montero et al., 2022).

We conduct experiments with different unsupervised and semi-supervised VAE-based disentangled representation learning methods in the well-known dSprites (Higgins et al., 2017), 3DShapes (Kim & Mnih, 2018), and MPI3D real (Gondal et al., 2019) datasets. Our experiments show the validity of our findings by showing that atomic group actions can be identified in disentangled models and used to implicitly cluster datasets. This demonstrates that the potential of synthesizing a full dataset, even from a limited or incomplete dataset, seems promising, which also may help increase the interpretability of cluster analysis and reduce storage space.

**The main contributions of this paper are**: (i) We introduce *Implicit Neural Clustering*, a simple sampling procedure that allows to implicitly cluster a dataset. (ii) Based on *Implicit Neural Clustering*, we provide a theoretical analysis of what happens when we equate clustering with disentangled representation learning, which leads to the discovery of a *lower bound* to clustering. (iii) We provide a practical implementation of *Implicit Neural Clustering* for disentangling VAEs and validate our approach through experiments on multiple datasets. (iv) We show an effective qualitative measure for disentanglement, which is more informative than the commonly-used disentanglement visualization with Hinton Matrices Eastwood & Williams (2018); Montero et al. (2022).

## 2 EQUATING CLUSTERING WITH DISENTANGLEMENT

Traditional explicit clustering is defined by a partition function $C_{sim}$ that partitions an input dataset $\mathcal{D}$, under an arbitrary notion of similarity $sim$, into $k$ clusters. $C_{sim}$ either maps any $x \in \mathcal{D}$ to a hard cluster assignment $C : \mathcal{D} \to \mathbb{N}$ (e.g., $k$-means) or soft cluster assignment $C : \mathcal{D} \to \mathbb{R}^k$ (e.g., maximum likelihood). Hard clustering can be defined as applying $C_{sim}$ on each $x \in \mathcal{D}$, which yields $k$ disjoint subsets $\mathcal{D}_k^{sim} \subseteq \mathcal{D}$ :

$$\mathcal{D} = \bigcup_k \mathcal{D}_k^{sim} = \bigcup_k \left\{ x \mid x \in \mathcal{D} \land C_{sim}(x) = k \right\} \tag{1}$$

with $\bigcup_k \mathcal{D}_k^{sim} = \mathcal{D}$ and $\bigcap_k \mathcal{D}_k^{sim} = \emptyset$. A different similarity $sim' \neq sim$ formally defines any arbitrary clustering different from $sim$ over $\mathcal{D}$, which is the basis for multi-partition clustering (Galimberti & Soffritti, 2007). In the multi-partition clustering context, explicit clustering w.r.t. $sim$

Figure 2: We visualize the difference between clustering in the traditional explicit sense and *Implicit Neural Clustering*. Under different measures of similarity, explicit clustering can cluster the dataset correctly in three different partitionings. However, since not all possible combinations between factors of variation are observed in the data, certain combinations are not present in the final clusters because we only *explicitly cluster* the real data. In contrast, *Implicit Neural Clustering* leads to *implicit clusters* that can also include novel cross-combinations not observed in the dataset.

yields only one out of many possible clusterings of the data. For multi-partition clustering, $sim$ can be considered in two ways. One, where $sim$ corresponds to clustering over different sub-dimensions of the feature representation, leading to different clustering partitions (Zhang, 2004; Galimberti & Soffritti, 2007; Vandewalle, 2020; Rodriguez-Sanchez et al., 2022; Falck et al., 2021; Willetts et al., 2019; de Chaumaray & Vandewalle, 2023). Two, where based on representation learning, one could train a different feature extractor for each possible $sim$.

## 2.1 Definition of Disentangled Representations

*Implicit Neural Clustering* builds on top of the established symmetry group-based definition for disentangled representations by Higgins et al. (2018). We briefly introduce essential parts needed from this definition. Let $G$ be a symmetry group acting on a set of world states (ground truth factors of variation) $W$, and let $O$ be a set of observations (e.g., pixel space) and $Z$ the internal agent representation of $W$. A generative process $b : W \to O$ leads from world to observation states, and an inference process $h : O \to Z$ leads from observation to an agent's internal representation of $W$. In this context, we have a dataset $\mathcal{D} = \{o_1, ..., o_N\}$ of observations $o_i \in O$. We now define the inference process $h : O \to Z$, as a parameterized feature extractor [1] $h_\varphi : O \to \mathcal{F}$ with parameters $\varphi$, which yields a disentangled representation $\mathcal{F}$ of any $o \in \mathcal{D}$. Under the assumption that $G$ can be decomposed into a direct product $G = G_1 \times ... \times G_M$, the representation $\mathcal{F}$ is disentangled with respect to $G$ provided that the following conditions are satisfied. (i) There are group actions that act on $\mathcal{F}$, $\cdot : G \times \mathcal{F} \to \mathcal{F}$, (ii) There is a mapping $d : W \to \mathcal{F}$, which is equivariant between the actions of $G$ on $W$ and $Z$: $g \cdot d(w) = d(g \cdot w), \forall g \in G, \forall w \in W$, and (iii) $\mathcal{F}$ decomposes into its factors of variation $\mathcal{F} = \mathcal{F}_1 \times ... \times \mathcal{F}_M$ so that any $\mathcal{F}_i$ is only affected by $G_i$ and invariant to any $G_j$, $\forall j \neq i$. Finally, we assume to have access to a parameterized generator $G_\theta : \mathcal{F} \to O$ with parameters $\theta$ that transforms samples from the disentangled representation space $\mathcal{F}$ to the observation space $O$.

## 2.2 From Explicit to Implicit Neural Clustering

In contrast to *explicitly* clustering a dataset $\mathcal{D}$ under various $sim$, *Implicit Neural Clustering* is derived from a *disentangled* representation $\mathcal{F}$ of $\mathcal{D}$. As an initial intuition, if we assume that any $o \in \mathcal{D}$ can be decomposed into its factors of variation, we can impose specific changes to any $o \in \mathcal{D}$ by modifying the desired parts of the factor in the representation. Figure 2 provides an overview of *Implicit Neural Clustering* with its main differences to *explicit* clustering. Following the definition of implicit probabilistic models, *Implicit Neural Clustering* can be defined as a sampling procedure from a disentangled latent space. Different from implicit models where a parameterized generator $G_\theta(\cdot)$ (e.g., GAN) transforms samples from an analytic distribution (e.g., isotropic Gaussian) to synthetic examples (Li & Malik, 2018), *Implicit Neural Clustering* transforms samples from a disentangled distribution $\mathcal{F}$ into synthetic clusters.

More specifically, for each cluster $\mathcal{D}_k^{sim}$, there exists an implicit cluster that can be obtained by sampling from $\mathcal{F}$ while fixing one respective factor of variation $\mathcal{F}_i$. Let $G = G_1 \times ... \times G_M$

---

[1] We change $Z$ to $\mathcal{F}$ for notation and readability reasons.

be the group actions that act on $\mathcal{F}$, and let $\cdot : G \times \mathcal{F} \rightarrow \mathcal{F}$ be the action that changes $\mathcal{F}$ to the respective factor $\mathcal{F}_i$. Given that each factor of variation has of several *atomic* attributes $\mathcal{F}_{ik}$ (e.g., the class labels of shape or color), we precisely define fixing a factor of variation as follows: Each $G_i$ consists of *atomic* partitions $G_i = \{G_{i1}, G_{i2}, ...\}$ that can modify any $z \in \mathcal{F}$ and are parameterized by a single value $f \in \mathbb{R}$ or a parameterized distribution $P(f \mid G_{ik})$. We further define a function $\overset{G_{ik}}{=}$ that yields "true" if an atomic factor of variation $\mathcal{F}_{ik}$ is present in $z \in \mathcal{F}$.

When clustering with respect to a factor of variation $\mathcal{F}_i$, let $sim \equiv \mathcal{F}_i$, equating disentanglement with clustering. Based on Equation 1 and the *atomic* partitions of a factor of variation, explicit clustering $C_{\mathcal{F}_i}$ splits $\mathcal{D}$ into $|G_i|$ disjoint subsets. In the *implicit* case, together with the generator $G_\theta$ and the feature extractor $h_\varphi$, we can generate each cluster $\mathcal{D}_k^{sim} \approx \mathcal{D}'_{if}$ implicitly to generate a synthetic version $\mathcal{D}'_i$ of the original dataset $D$.

$$\mathcal{D} \approx \mathcal{D}'_i = \bigcup_{f \in G_i} \mathcal{D}'_{if} = \bigcup_{f \in G_i} \{G_\theta(h_\varphi(o)) \mid o \in \mathcal{D} \wedge \overset{f}{=} (h_\varphi(o))\} \tag{2}$$

We have that $\mathcal{D}'_i$ implicitly models $\mathcal{D}$ with respect to a clustering under a factor of variation $\mathcal{F}_i$, under the assumption that the encoder $h_\varphi$ is capable of disentangling and $G_\theta$ is capable of realistically reconstructing the encoded elements. More specifically, $h_\varphi$ must disentangle any $o \in \mathcal{D}$ w.r.t. $\mathcal{F}_i$, so that $sim \equiv \mathcal{F}_i$, and $G_\theta$ must recover an $o'$ from this representation so that $o' \approx G_\theta(h_\varphi(o))$. To move beyond *explicit* clusters, we further assume the disentangled representation space $\mathcal{F}$ to be composable, i.e., we can modify any $z \in \mathcal{F}$ by acting with the atomic group action $G_{if}$, which changes cluster membership under factor $\mathcal{F}_i$ from any previous $D'_{il}$ to $D'_{if}, l \neq f$, or produce a variation of any $o \in \mathcal{D}$ by acting with the atomic group action $G_{if}$ of the same cluster on $z$. Together with the compositionality assumption, Equation 2, the disentanglement assumption on $h_\varphi$, and the reconstruction assumption on $G_\theta$, we derive *Implicit Neural Clustering*.

$$\mathcal{D} <\approx \mathcal{D}'_i = \bigcup_{f \in G_i} \{G_\theta(z) \mid z \in \{\cdot(f, z_1), ..., \cdot(f, z_K)\} \sim \mathcal{F}\} \tag{3}$$

where $(\cdot) \sim \mathcal{F}$ denotes a sampling procedure for each factor of variation $\mathcal{F}_i$ and $K$ denotes the number of elements to be sampled. For *Implicit Neural Clustering* we fix a factor of variation $\mathcal{F}_i$, sample latent representations $z \in \mathcal{F}$ using the sampling procedure $(\cdot) \sim \mathcal{F}$, take/sample a respective group action[2] $f$ of an atomic factor of variation $\mathcal{F}_i$, and modify each $z$ accordingly with $\cdot(f, z)$. Up to the capabilities of the encoder $h_\varphi$ and generator $G_\theta$, $\mathcal{D} <\approx \mathcal{D}'_i$ emphasizes that the set $\mathcal{D}'_i$, obtainable from the outlined procedure, can at least *implicitly* represent the original dataset $\mathcal{D}$ as a *lower bound*. The lower bound[3] to Equation 3 is given by the reconstruction of the dataset in Equation 2. Therefore, up to the capabilities of the encoder $h_\varphi$ and generator $G_\theta$, *Implicit Neural Clustering* is able to (a) generate a variety of realistic data by sampling arbitrary data compositions and (b) synthesize novel examples in each cluster not observed in the dataset when $G_\theta$ is capable of combinatorial generalization. Under the respective definition of disentangled representations in Section 2.1 and Equations 1, 2, and 3, the resulting synthetic dataset $\mathcal{D}'_i$ is partitioned into disjoint subsets w.r.t. a fixed factor of variation $\mathcal{F}_i \equiv sim$. In this way, we define an implicit clustering of $\mathcal{D}$, where clustering is equated with disentanglement, and clusters are generated implicitly by a generative model controllable by disentangled factors of variation.

**Obtaining atomic group actions in disentangling VAEs.** We provide a simple procedure to identify atomic group actions in the latent space of disentangling VAEs, which often encode factors of variation in only one dimension $l$ of the representation $z = (z_1, z_2, ..., z_d) \in \mathbb{R}^d$, $d > M$. To obtain the atomic group actions, we first encode the full dataset and then partition each dimension using kernel density estimation (KDE) at local minima of the resulting density estimates[4]. This leads to density-based partitions, which naturally arise in the latent space of disentangling VAEs and each partition is a parameterized probability distribution $P(f|G_{ik})$ (e.g., uniform or normal),

---

[2]In practice, we would parameterize $f$ with a probability distribution and sample the respective modification for more variety, but a single value, like the mean over all possible values, would also work.

[3]Extending on this proof sketch, we provide a proof in Appendix A1

[4]Any partition algorithm could be used. KDE has the advantage over, e.g., $k$-means that we do not specify the number of partitions in advance.

which resembles a probability distribution over atomic group actions for $\mathcal{F}_{ik}$ that we can sample from. However, it is important to point out that these simple partitions are only meaningful when a factor of variation is properly disentangled.

**Sampling Procedure for Implicit Neural Clustering.**

Given atomic group actions, the specific sampling procedure *Implicit Neural Clustering* is defined in lines 5-8 in Algorithm 1. First, we sample a random value from the partition distribution $f \sim P(f|G_{ik})$. Next, we sample a random latent $z \in \mathcal{F}$.

This process can be done in two ways. (1) Sample only from the found partitions (AS) or (2) sample $z$ from the set of all encoded data-points $h_\varphi(\mathcal{D})$ ($\neg$AS). Afterward, we act with $f$ on $z$, i.e., $\cdot(f, z)$, which modifies $z$ accordingly. Acting can also occur with multiple actions at the same time (e.g., when a factor is disentangled across two dimensions). In the latter case, we denote $\cdot()$ as Multi-Dimensional Action (MDA). Finally, the process in lines 5-8 is repeated $K$ times for each cluster $\mathcal{D}_{ik}$. Note that it is straightforward to achieve an implicit multi-partition clustering by simply repeating Algorithm 1 with different $\mathcal{F}_i$.

**Input:** Group actions $G_i$ for factor of variation $\mathcal{F}_i$,
Generator $G_\theta$, number of samples $K$
**Output:** Implicit clustering $\mathcal{D}'$ with respect to $\mathcal{F}_i$

1   $\mathcal{D}'_i \leftarrow \emptyset$
2   **for** $k$ *in* $1..|G_i|$ **do**
3     $D_{ik} \leftarrow \emptyset$
4     **for** $1..K$ **do**
5       $f \sim P(f|G_{ik})$
6       $z \sim AS(\mathcal{F})$ or $\neg AS(\mathcal{F})$
7       $z' = \cdot(f, z)$ or $MDA(f, z)$
8       $D'_{ik} \leftarrow D'_{ik} \cup \{G_\theta(z')\}$
9     **end**
10    $D'_i \leftarrow D'_i \cup D'_{ik}$
11  **end**

**Algorithm 1:** Our sampling procedure

**Identifying meaningful atomic group actions in disentangling VAEs.** When ground truth factors of variation are available, we can identify meaningful disentangled atomic group actions by computing if they "uniquely" co-occur with known ground truth atomic factors of variation $\mathcal{F}_{ij}$. To this end, we count the frequency of co-occurences between all extracted KDE partitions $pt_a$, $a \in \mathbb{N}$ and atomic ground truth factors of variation $\mathcal{F}_{ij}$ of each $o \in \mathcal{D}$. This leads to matrix with the factors of variation as rows and the partitions as columns. For each cell $[(i, j), a]$ in row $(i, j)$, we divide the frequency $freq(\mathcal{F}_{ij}, pt_a)$ by the sum over all frequencies of the row $(i, j)$, which leads to: $[(i, j), a] = \dfrac{freq(\mathcal{F}_{ij}, pt_a)}{\sum_b freq(\mathcal{F}_{ib}, pt_b)}$. Using this method, we can visualize disentanglement in a qualitative manner as shown in Figure 3, where unique co-occurences between a partition and a factor of variation show meaningful disentanglement.

To identify meaningful atomic group actions in the unsupervised case, where no information about ground truth factors of variation is available, one can fall back (i) to a general-purpose zero-shot classifier like CLIP (Radford et al., 2021) to predict if a factor of variation is consistently present in batch of randomly modified images with the same KDE partition, or (ii) use general-purpose embeddings from, e.g., DINOv2 (Oquab et al., 2023) to find a significant image similarity increase when a batch of random images is modified with the same KDE partition.

## 3   EXPERIMENTAL EVALUATION

In this evaluation, we systematically increase the number of assumptions made on the encoder $h_\varphi$ and generator $G_\varphi$ that must be fulfilled for *Implicit Neural Clustering* to be valid. In the first set of experiments, we evaluate *Implicit Neural Clustering* with the assumption $A.H$ that an encoder $h_\varphi$ disentangles the factors of variation in atomic partitions. In this setting, we propose an interpretable procedure to quantify and assess qualitatively when $A.H$ is satisfied. Furthermore, for cases where $A.H$ is satisfied, we quantitatively and qualitatively evaluate the quality of the generated implicit clusters, showing that the assumptions made on the generator $G_\varphi$ regarding realistic sample generation ($A.G1$) and compositionality of atomic partitions for controllable synthesis ($A.G2$) are valid. Afterward, we evaluate the partition performance using KDE and compare it against traditional clustering algorithms, empirically validating the *lower bound* to *Implicit Neural Clustering*. Furthermore, we perform an ablation study on the sampling procedure regarding sampling type AS and acting type MDA to show the impact of disentanglement on the overall performance. Finally, our

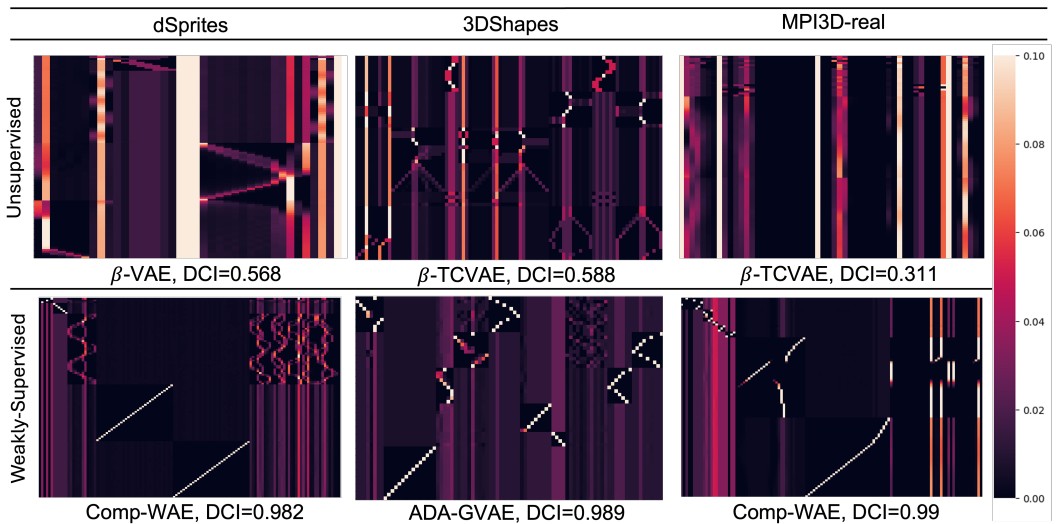

Figure 4: Co-occurence plot between atomic factors of variation and the dimension-wise partitions obtained with KDE.

last experiment discusses the validity, strengths, and limitations of *Implicit Neural Clustering* under combinatorial generalization (assumption $A.G3$) of $G_\theta(\cdot)$ in light of the results of existing works.

**Experimental Setup.** To show the validity and limitations of our concept, we consider 3DShapes (Kim & Mnih, 2018), MPI3D Real (Gondal et al., 2019), and dSprites (Higgins et al., 2017), which are widely adopted datasets for disentangled representation learning. Specifically, we evaluate learning disentangled representations in an unsupervised setting with $\beta$-TCVAE (Chen et al., 2018), and in a weakly-supervised setting with ADA-GVAE (Locatello et al., 2020) and the architecture of (Montero et al., 2020; 2022), which we refer to by Comp-WAE. We use these models as a means to show the validity of our concept due their strong disentanglement performance w.r.t. the DCI disentanglement metric (Eastwood & Williams, 2018) and their shown compositionality capabilities. The DCI disentanglement metric measures the degree of capturing at most one generative factor for each latent variable. In all experiments[5], all models are trained from scratch on a single 46GB RTX A6000.

**E0 w/ $A.H$: Qualitatively probing the applicability of *Implicit Neural Clustering*.**
We propose a simple but effective qualitative evaluation procedure for our very strict disentanglement requirement into atomic partitions. As presented in Figure 3, we propose a visualization scheme that is more informative than Hinton Matrices eastwood2018framework, montero2022lost). Compared to a Hinton Matrix (left), our method provides much more details on disentanglement without the need for a classifier, which makes it a complementary visualization tool to Hinton Matrices and DCI (proposed in (Eastwood & Williams, 2018)) for assessing disentanglement. Figure 4 compares several unsupervised models with

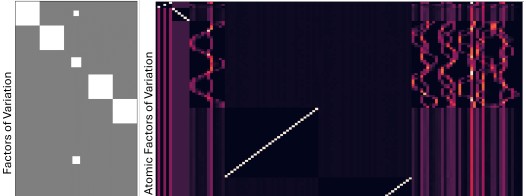

Figure 3: Comparing a Hinton Matrix against atomic group actions. Atomic group actions in VAEs are well disentangled and can be used to control the generation.

weakly-supervised approaches under DCI and shows the corresponding qualitative co-occurrence plot between found atomic partitions and ground truth factors of variation. We find overall that we are always able to consistently generate the implicit clusters based on factors of variation that can be seen in Figure 4 as clean and unique partitions. Furthermore, a lower DCI score indicates worse disentanglement that correlates with our qualitative measure.

---

[5]Code, models, and all details to reproduce our experiments will be publicly available upon acceptance.

Table 1: Quality of generated samples in different datasets with different models.

| Dataset | Approach | DCI | $\mathcal{F}_1$ | $\mathcal{F}_2$ | $\mathcal{F}_3$ | $\mathcal{F}_4$ | $\mathcal{F}_5$ | $\mathcal{F}_6$ | $\mathcal{F}_7$ |
|---------|----------|-----|-----|-----|-----|-----|-----|-----|-----|
| | | | | | *F*1 Macro (↑) | | | | |
| dSprites | CompWAE | 0.999 | 0.422 | 0.945 | 0.118 | 0.619 | 0.698 | - | - |
| | Oracle | - | 0.99 | 0.99 | 0.92 | 0.65 | 0.70 | - | - |
| 3DShapes | ADA-GVAE | 0.99 | 0.73 | 0.64 | 0.46 | 0.95 | 0.96 | 0.98 | - |
| | Oracle | - | 1.0 | 1.0 | 1.0 | 1.0 | 1.0 | 1.0 | - |
| MPI3D | CompWAE | 0.99 | 1.0 | 0.49 | 0.98 | 1.0 | 1.0 | 0.28 | 0.85 |
| | Oracle | - | 1.0 | 0.98 | 0.99 | 1.0 | 1.0 | 0.90 | 0.99 |

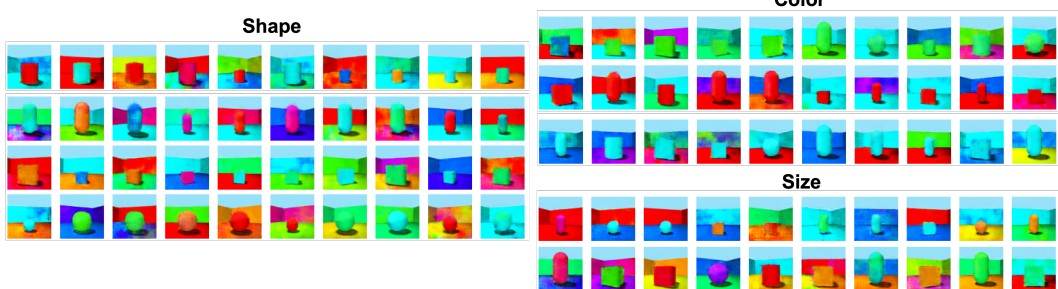

Figure 5: Exemplary *Implicit Neural Clustering* of 3DShapes. Each row represents random samples for some atomic factor of variation. Each row is the result of applying atomic group actions we extracted from the disentangled representation space to random samples.

From experiment E0, we find that unsupervised methods do not satisfy the necessary disentanglement requirement for our theoretical assumptions. For this reason, the remaining experiments of this section are performed only with weakly-supervised approaches, showing the validity of our theoretical findings.

**E1 w/ $A.H$, $A.G1$, $A.G2$: Quantifying Realism of the Generated Samples.** To evaluate the quality of the generated samples, we train a CNN as an oracle on the images of the real dataset to predict which factor of variation is present in each image. For each $\mathcal{F}_i$, we (1) split the dataset into training and test sets (random 0.8/0.2), (2) train on the training split of the real data for 15 epochs to predict $\mathcal{F}_i$, (3) generate an *Implicit Neural Clustering* of the dataset w.r.t. a factor of variation $\mathcal{F}_i$ with 10,000 samples for each atomic factor, and (4) evaluate the classifier on the real test split as well as on the synthetically generated dataset. In step (3), we run the sampling procedure from Algorithm 1 with $MDA$ and $\neg AS$ as components. We found this combination to perform the best (see ablation study in Experiment E4). The results are shown in Table 1, where we report the macro $F1$ score over all atomic factors of each $\mathcal{F}_i$. For very low scores, some implicit clusters can be generated well while others fail (full confusion matrices are provided in the supplementary material). We notice that MPI3D and Shapes3D can be implicitly clustered much better than dSprites. To summarize, most of the generated samples can be predicted accurately compared to the real data, showing that the generated samples are (1) realistic because there is only small drop in performance on the oracle performance, and (2) can be synthesized by acting with the atomic partitions, showing the assumption of compositionality is fulfilled. The quantitative results relate to the qualitative evaluation procedure in Figure 4, in which the factors of variation that are "atomically" disentangled provide the highest $F1$ scores. Analogously, the factors that do not have unique co-occurences with atomic partitions or that span multiple dimensions exhibit lower quality in the generated samples.

**E2 w/ $A.H$, $A.G1$, $A.G2$: Qualitative Evaluation of Implicit Neural Clustering.** Figure 5 shows three coherent synthetic multi-partition clusters of 3DShapes with respect to shape, color, and size that we have *implicitly* clustered with our concept. We obtained these results by applying to random samples the atomic group actions that we were able to extract from the disentangled representations of the underlying dataset. In Figure 6 we present the modifications of arbitrary samples based on the atomic group actions, which demonstrates that randomly generated samples can be modified to the desired atomic factor the variation. It is relevant to note that although we are able to find atomic group actions for all ground truth labels, not all of them are invariant to the other factors of variation, which demonstrates a limitation in the disentanglement for some factors of variation.

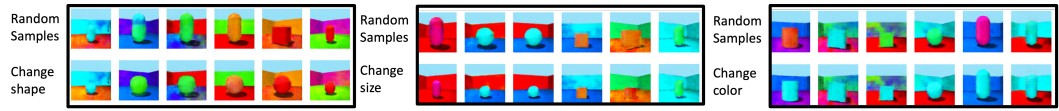

Figure 6: For random generated samples, atomic group actions specifically modify a certain factor of variation, e.g., change the object shape, color, or size. Exemplary demonstration in 3DShapes.

Table 2: Comparing partitioning performance of KDE against $k$-Means and GMM.

| ACC,NMI ($\uparrow$) | $\mathcal{F}_4$ | | $\mathcal{F}_5$ | | $\mathcal{F}_6$ | |
|---|---|---|---|---|---|---|
| $k$-means (requires number of partitions) | 0.87 | 0.92 | 0.75 | 0.80 | 1.0 | 1.0 |
| GMM (requires number of partitions) | 0.87 | 0.92 | 0.75 | 0.82 | 0.99 | 0.99 |
| KDE (non-parametric) | **0.99** | **0.97** | **1.0** | **1.0** | **0.99** | **0.99** |

In the supplementary material, we provide the full implicit clusters for all three datasets and latent traversals with atomic group actions for MPI3D and 3DShapes.

**E3: On Clustering Performance and the Lower Bound.** Using the 3DShapes and the best-performing factors from Table 1, we compare KDE against $k$-means and a Gaussian Mixture Model (GMM) to show its effectiveness in partitioning atomic group actions in each dimension. At the same time, this experiment allows to evaluate the clustering performance and empirically validate the *lower bound* to clustering given by Equation 2. When a factor of variation is used as the measure of similarity, then under a high degree of disentanglement, implicit clusters are equivalent to explicit clustering. We set the number of clusters in $k$-means and GMM to the ground truth number of atomic factors corresponding to each dimension, which we identify with our qualitative co-occurrence disentanglement measure. KDE does not require any number of clusters as a parameter. We use the commonly used Purity (ACC) and Normalized Mutual Information (NMI) as metrics. The results are shown in Table 2, where we can see that using KDE to partition a dimension outperforms both $k$-Means and GMM in all factors. In addition, another advantage of using our KDE approach is that it is non-paramatric, i.e., does not require specifying the number of clusters in advance. Together with the realism of generated samples in Table 1, the implicit clusters in Figure 5, and the partitioning performance in Table 2, it becomes evident that we can implicitly cluster a dataset by controllable atomic factors of variation. That is, because (i) implicit clusters in 2 are equivalent to explicit clustering due to reconstruction (Equation 2), and (ii) randomly generated samples for each cluster always include the factor of variation (Equation 3). These results empirically show the *lower bound* for clustering.

**E4: Ablation Study, Sampling Procedure.** We test two different kinds of sampling strategies using the 3DShapes dataset. First, sampling only from the atomic partitions (AS) or using random samples from the encoded dataset ($\neg$AS). Second, acting with multi-dimensional group actions (MDA) or only acting with a single group action ($\neg$ MDA). Table 3 shows the results for each combination. We can see that $\neg$AS together with MDA lead to the best results. It is expected for AS to exhibit low performance, since not all factors factors of variation are perfectly disentangled. Given the qualitative disentanglement results, it is also expected that MDA should perform better, since some factors are disentangled across multiple dimensions while still being unique combinations. This result highlights that compositionality can even apply to the tested models when atomic actions span multiple dimensions at the same time.

**Note on our Results and Combinatorial Generalization (w/ $A.H$, $A.G1$, $A.G2$, $A.G3$).** Among other works, Montero et al. (Montero et al., 2022) have shown that combinatorial generalization can be achieved in some special cases with learned disentangled representations. Despite these special cases, there is no theoretical guarantee that proper disentanglement leads to combinatorial generalization or that these special cases will transfer across different kinds of models. Dividing the problem in two, i.e., disentangling first, and training a separate generator on the disentangled representations afterwards potentially leads to better results. Building on the theory of our concept and the empirical results of our experiments, we can straightforwardly apply our concept in a setting where $G_\theta$ has learned combinatorial generalization ($A.G3$). In this setting, our experiments close the loop in the conceptual illustration of our concept in Figure 2, where $G_\theta$ will "fill-in-the-blanks" by synthesizing cross-combinations between factors of variation not seen in the dataset. In the same

Table 3: Ablation study regarding the impact of sampling procedure on the performance of the generated implicit clusters. We evaluate predicting the factor of variation using classifiers trained on the real data in all generated implicit clusters.

| | | $F1$ Macro ($\uparrow$) | | | | | |
|---|---|---|---|---|---|---|---|
| w/ AS | w/ MDA | $\mathcal{F}_1$ | $\mathcal{F}_2$ | $\mathcal{F}_3$ | $\mathcal{F}_4$ | $\mathcal{F}_5$ | $\mathcal{F}_6$ |
| | | 0.43 | 0.39 | 0.52 | 0.88 | 0.98 | 0.98 |
| | $\checkmark$ | **0.73** | **0.64** | **0.46** | **0.95** | **0.96** | **0.98** |
| $\checkmark$ | | 0.36 | 0.44 | 0.20 | 0.74 | 0.91 | 0.81 |
| $\checkmark$ | $\checkmark$ | 0.68 | 0.56 | 0.38 | 0.74 | 0.95 | 0.79 |
| Ground Truth $\mathcal{D}$ | | 1.0 | 1.0 | 1.0 | 1.0 | 1.0 | 1.0 |

context, our concept can be applied to any kind of models trained on ground truth factors of variation that have learned combinatorial generalization (e.g., with conditional or composable diffusion models (Okawa et al., 2024; Liu et al., 2022)) to implicitly cluster the data. Further experiments to validate the aforementioned claims are provided in Appendix C and Appendix D. However, even in this relaxed setting, where the ground truth factors of variation are known in advance (which resembles an optimal encoder $h^*$), there is still no theoretical guarantee for combinatorial generalization in $G_\theta$, when trained on the factors of variation obtained by an optimal encoder $h^*$. Therefore, in line with many previous works (e.g., (Montero et al., 2020; 2022; Okawa et al., 2024; Wiedemer et al., 2024b), it is important to emphasize that disentanglement and compositionality does not imply combinatorial generalization. We stress that our concept is constrained by the current limitations faced by the field in relation to combinatorial generalization.

## 4 DISCUSSION

Our experiments show that *Implicit Neural Clustering* has a *lower bound* and is particularly limited by the assumptions on $h_\varphi$ regarding disentanglement ($A.H$), and $G_\theta(\cdot)$ regarding realism of synthetic samples ($A.G1$), compositionality ($A.G2$), and combinatorial generalization ($A.G3$). Especially for disentanglement, we notice a huge gap between unsupervised and weakly-supervised approaches. While combinatorial generalization is feasible in relaxed synthetic settings and to some extent in real-world data (Wiedemer et al., 2024a; Montero et al., 2022; Okawa et al., 2024), effective methods for learning disentangled representations and achieving combinatorial generalization from complex real-world data remain elusive. In real-world tasks, general-purpose embeddings learned through SSL methods like SimCLR (Chen et al., 2020) or DINOv2 (Caron et al., 2021; Oquab et al., 2023) can learn effective representations that disentangle real-world data to limited extent. Empirical evidence of disentanglement in SSL representations, as shown by Bordes et al. (2022), demonstrates that training a generator on SSL features allows for concept swapping in representations, producing samples that reflect these changes. While our approach naturally extends to SSL representations and would make it applicable to datasets like CIFAR-10 (Krizhevsky et al., 2009) or ImageNet (Deng et al., 2009), atomic group actions do not exist in the simple form as in disentangling VAEs. New methods to identify potential subspaces in these representations may lead to new insights. A different way to apply our method on real-world data would be through finding a way that effectively partitions the interpretable directions learned by latent navigators from GANs (e.g., (Voynov & Babenko, 2020; Georgopoulos et al., 2022)), or Diffusion Models (e.g., (Yang et al., 2023)). Finally, it is easy to see that our sampling procedure can be easily applied to generative models trained on ground truth factors of variation to synthesize datasets. We point to an important closely related work by (Okawa et al., 2024), where a conditional diffusion model is trained on ground truth factors of variation and combinatorial generalization is achieved. In this context, our work provides valuable insights on a research question by Jahanian et al. (2021): "If we have good enough generative models, do we still need datasets?" With *Implicit Neural Clustering*, we can potentially generate any realistic synthetic variations of a dataset with a corresponding class label, fill gaps in its distribution, and it can be a basis to replace datasets in order to save valuable cost for storage and acquisition of data. Applications of *Implicit Neural Clustering* are not only limited to datasets of images but can also be applied to completely different kinds of data, such as natural language, videos, or time series. Negative societal impact can occur when a model can achieve combinatorial generalization under "full" disentanglement for, e.g, DeepFakes in the imaging or video domain.

## 5 RELATED WORK

Existing *related works* on disentangled representation learning, deep generative clustering, and controllable image generation have shown the following points. (i) Factors of variation are embedded in single (e.g., (Locatello et al., 2019; 2020; Wang et al., 2022)) or multiple dimensions (e.g., (Bordes et al., 2022; Falck et al., 2021)) of a disentangled latent space, which can be learned with disentangled representation learning approaches that are unsupervised (e.g., VAE-based (Higgins et al., 2017; Kim & Mnih, 2018; Locatello et al., 2019; Falck et al., 2021), from pre-trained generative models (Ren et al., 2022; Yang et al., 2023), deep-clustering (Mukherjee et al., 2019; Lee et al., 2020; Yu & Welch, 2021; Ding et al., 2022; Zhao et al., 2020)), or (weakly) supervised (e.g., (Hristov et al., 2018; Locatello et al., 2020; Montero et al., 2020; 2022; Wang et al., 2022)), (ii) Deep generative clustering approaches (e.g., (Mukherjee et al., 2019; Lee et al., 2020; Yu & Welch, 2021; Ding et al., 2022)) can simultaneously learn a disentangled latent space, cluster assignments, and allows controllable generation of elements for each cluster with disentanglement. (iii) Learned disentangled representations are often composable and can be used to control the factors of variation in images using latent traversal or the recombination/swapping of different latent dimensions between images (e.g., (Bordes et al., 2022; Wang et al., 2022; Falck et al., 2021; Montero et al., 2020; 2022)). (iv) Both generative and disentangled representation learning models can learn combinatorial generalization in rare synthetic and real-word settings (e.g., (Okawa et al., 2024; Montero et al., 2020; 2022; Wiedemer et al., 2024b)), which allows to synthesize novel cross-combinations between factors of variation not observed in the data. (v) Depending on the degree of disentanglement, meaningful directions to traverse a disentangled latent space to control images can be straightforward one-dimensional and linear (e.g., (Higgins et al., 2017; 2018)), or non-linear multi-dimensional traversals can be learned from disentangled latent spaces in VAEs (e.g., (Ren et al., 2022; Yang et al., 2023)), GANs (e.g., (Voynov & Babenko, 2020; Georgopoulos et al., 2022)), or Diffusion Models (e.g., (Yang et al., 2023)) in the form of a navigator.

Based on the points above, existing methods can already effectively control and sample data for implicit cluster generation. However, they primarily focus on improving clustering performance, disentanglement performance, controllable generation, realism, or combinatorial generalization in isolation. Instead of treating these problems individually, with *Implicit Neural Clustering*, our work takes a new unified perspective by equating disentanglement with clustering and generative models. This reveals that existing methods are inherently limited by a clustering *lower bound*, given by disentanglement capabilities, realism of generated samples, and combinatorial generalization.

## 6 CONCLUSION

In this paper, we present *Implicit Neural Clustering*, a sampling method for generating clusters implicitly through disentangled representations. Through theoretical analysis and empirical validation, we show that equating disentangled representation learning with clustering and generative models reveals that this method has *lower bound*, governed by the degree of disentanglement, realism of generated samples, and combinatorial generalization in generative models. This *lower bound* of *Implicit Neural Clustering* highlights strong potential for relevant future applications, such as implicitly generating clusters driven by factors of variation in real datasets, synthesizing complete datasets from limited data, improving interpretability in cluster analysis, enhancing SSL and classification tasks, and reducing data storage needs. At last, in line with many prior works, we also underscore the importance of focusing on combinatorial generalization in future research.

REPRODUCIBILITY STATEMENT

We commit to ensuring the reproducibility of our work as follows:

- All of our implementations, trained model checkpoints, hyperparameters, and all necessary details to reproduce our empirical and qualitative results will be publicly available upon acceptance.
- The source code of models by related work and the datasets used in this work are all publicly available.
- The hardware setup used for our experiments (46GB NVIDIA RTX A6000 GPU) is described.

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

APPENDICES

## A    PROOF FOR LOWER BOUND

We now provide a proof that Equation 2 is a lower bound for implicit clustering, which is given by explicit clustering.

*Proof.* Suppose we cluster $\mathcal{D}$ by $sim$, then we obtain $k$ disjoint subsets of $\mathcal{D}$ (Equation 1). When $sim \equiv \mathcal{F}_i$ holds, Equation 2 formalizes the generation of any partitioning of the data. Under an encoder that perfectly disentangles any datapoint into its factors of variation in single dimensions, we can transform any $o \in \mathcal{D}$ to a latent representation that include its factors of variation $z = h_\varphi(o)$. Given that the factors are perfectly disentangled, we can then group all $o \in \mathcal{D}$ by the respective dimension representing the respective factor of variation $\mathcal{F}_i$ and obtain the same disjoint subsets as with explicit clustering under $C_{sim}$. If instead of explicitly grouping the elements $o$, we reconstruct them using a generator $G_\theta(\cdot)$, i.e., $o^* = G_\theta(h_\varphi(o))$, we specifically end up with Equation 2:

$$\mathcal{D} = \bigcup_k \mathcal{D}_k^{sim} = \bigcup_k \{x \mid x \in \mathcal{D} \wedge C_{sim}(x) = k\} = \bigcup_{f \in G_i} \{o \mid o \in \mathcal{D} \wedge \overset{f}{=} h_\varphi(o)\} \quad (4)$$

$$\approx \bigcup_{f \in G_i} \{G_\theta(h_\varphi(o)) \mid o \in \mathcal{D} \wedge \overset{f}{=} h_\varphi(o)\} = \bigcup_{f \in G_i} \mathcal{D}'_{if} = \mathcal{D}' \approx \mathcal{D} \quad (5)$$

Up to the reconstruction capabilities of the generator $G_\theta$, the reconstructed dataset $\mathcal{D}'$ is a synthetic version of $\mathcal{D}$, which under a perfect generator would be equivalent, i.e., $\mathcal{D} \equiv \mathcal{D}'$. However, because we can not assume a perfect reconstruction from the latent representation $z$, we write the synthetic version of $\mathcal{D}'$ is an appropriate realistic synthetic version of $\mathcal{D}$, i.e., $\mathcal{D} \approx \mathcal{D}'$.

Suppose that we can modify the factors of variation of an object $o$ in its disentangled representation $z = h_\varphi(o)$ with a function $mod$, and the generator $G_\theta$ creates realistic synthetic elements $o' = G_\theta(mod(h_\varphi(o)))$. In this case, any minor modification yields a new object $o'$ different from $o$, effectively extending the cardinality and variety of samples in $\mathcal{D}'$, which makes Equation 2 a lower bound to *Implicit Neural Clustering*, given by explicit clustering. More specifically:

$$\mathcal{D} = \bigcup_k \mathcal{D}_k^{sim} <\approx \bigcup_{f \in G_i} \mathcal{D}'_{if} = \mathcal{D}' \quad (6)$$

$$= \bigcup_{f \in G_i} \{G_\theta(h_\varphi(o)) \mid o \in \mathcal{D} \wedge \overset{f}{=} h_\varphi(o)\} \cup \{G_\theta(mod(h_\varphi(o))) \mid o \in \mathcal{D} \wedge \overset{f}{=} mod(h_\varphi(o))\} \quad (7)$$

Assuming the factors of variation $F$ for any $x \in \mathcal{D}$ can be obtained with an encoder $h$, derived analytically, or are given by annotations, we can train $G_\theta(\cdot)$ to synthesize samples from the underlying distribution that we can group naturally by a factor of variation. A more general formulation that encompasses compositional and combinatorial generalization is given in Equation 3. When we can partition or factorize the underlying generative factors into respective atomic partitions and compositionality emerges in $G_\theta(\cdot)$, synthesis of known cross combinations between factors of variation is possible, which results in (a) dataset reconstruction (Equation 2) and (b) controllable synthesis, i.e., we can now move beyond only reconstructing the dataset, but can also specifically control the synthesis of new examples. Finally, when $G_\theta(\cdot)$ also learns to generalize to combinations that are not in the data distribution, we move beyond the lower bound given by Equation 2 with Equation 3, where arbitrary novel cross-combinations between factors of variation can be synthesized.  □

## B    ADDITIONAL DETAILS FOR EXPERIMENTAL SETUP AND DESIGN

We will provide additional training details on hyperparameters, code, and further setups upon acceptance of this publication.

The abbreviations for each factor of variation $\mathcal{F}_i$ in each dataset as used in our experiments is based on the specification of the datasets.

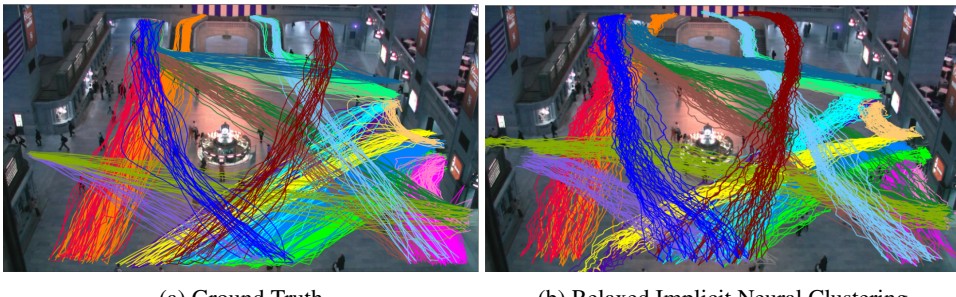

(a) Ground Truth        (b) Relaxed Implicit Neural Clustering

Figure 7: Relaxed implicit neural clustering of crowd trajectory data (right) effectively approximates the underlying trajectory distribution (left)

## C    APPLICATION OF IMPLICIT NEURAL CLUSTERING TO OTHER DOMAINS

Multi-partition clusters exist in many different domains. Different from the hard challenge of learning the factors of variation with an encoder $h_\varphi$, they are often provided in ground truth or can be obtained analytically. In such contexts, where factors of variation might be already available, can be obtained analytically, or with a zero-shot classifier like CLIP, *Implicit Neural Clustering* is also applicable when we relax the assumption on having an encoder $h_\varphi$. To show this, in the following experiment, we relax the assumption that there exists an encoder $h_\varphi$ that extracts factors of variation from a given dataset $\mathcal{D}$ to show that the compositionality and combinatorial generalization requirements for our approach can be fulfilled in different applications. More specifically, we assume that factors of variation have been obtained by, e.g., applying an "optimal" encoder $h^*$ (e.g., a human, an analytical relaxation, or a zero-shot classifier Radford et al. (2021)) on $\mathcal{D}$. In such cases, *Implicit Neural Clustering* is applicable to generate implicit clusters resembling the conceptual overview of our approach in Figure 2. While this relaxation might seem trivial, we can effectively show a novel application of disentangled representations, while also providing a new perspective on clustering.

**Experimental Setup and Models.**    To show different applications of *Implicit Neural Clustering* with analytical or provided ground truth factors of variation and different domains, we evaluate our approach with the following datasets. (i) the Grand Central Station (GC) Zhou et al. (2012) dataset that consists of time series that resemble trajectories of pedestrians traversing a public train station. In GC, we can vary the start and goal position of the agent to traverse the underlying environment, which passed to a generator as continuous inputs. In this dataset, we train a goal-conditioned policy with behavior cloning as the generator $G_\theta$ Kreutz et al. (2024). (ii) the CLEVR Relations Johnson et al. (2017) dataset, which allows us to vary three ground truth factors of variation, the number of objects, as well as their respective X and Y coordinates. We use a pre-trained compositional diffusion model Liu et al. (2022) provided by the authors[6] that is conditioned on natural language prompts. In this way, we show an application to text-to-image generative models. (iii) a synthetic dataset of simple shapes provided by Okawa et al. (2024) (in the remainder referred to as Simple-Shapes). SimpleShapes has shape, color, and size as factors of variation, which are used as continuous input to the generator. We train a conditional diffusion model [7] based on Okawa et al. (2024) in SimpleShapes on the provided ground truth factors of variation. All models are trained from scratch with a single 46GB RTX A6000. We will provide additional training details on hyperparameters, code, and further setups upon acceptance of this publication.

### C.1    ANALYTICAL FACTORS OF VARIATION — SEQUENTIAL DECISION MAKING.

Sequential decision making tasks, such as motion planning or navigation, can be relaxed analytically into several factors of variation. For instance, start and goal positions influence the outcome of a trajectory. When varying these two positions as factors of variation, a policy trained on a dataset of expert demonstrations will then "fill-in-the-blanks" and generate a trajectory that follows the

---

[6]https://energy-based-model.github.io/Compositional-Visual-Generation-with-Composable-Diffusion-Models/

[7]public github repository `https://github.com/phys-ai/concept_graphs/$`

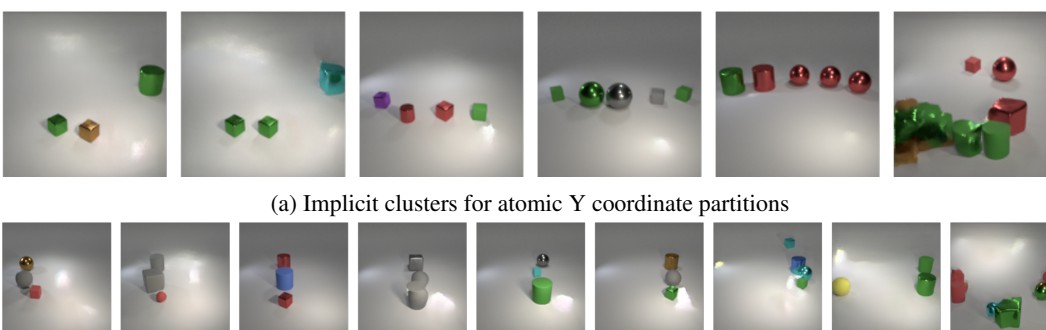

(a) Implicit clusters for atomic Y coordinate partitions

(b) Implicit clusters for each atomic X coordinate partitions

Figure 8: Relaxed implicit neural clustering of object compositions, where one can implicitly cluster the data based on the relaxed representation of the factors of variation, i.e., discretized x and y coordinates

ground truth distribution. In this setting, we can apply *Implicit Neural Clustering* where start and goal positions can be considered factors of variation.

We evaluate *Implicit Neural Clustering* on the grand central station dataset (GC) Zhou et al. (2012), where we train a goal-conditioned policy with behavior cloning as the generator $G_\theta$. Analogous to images, we partition the factors of variation in each dimension separately, i.e., start and goal positions using DBSCAN as a partition algorithm, and only keep the top $k$ number of (start, goal) pairs from the real dataset as factors of variation. Given the respective positions at each start and goal, we compute the parameters of a normal distribution for each of these sets, which serve as the parameters to sample from atomic start and goal partitions for *Implicit Neural Clustering*.

Figure 7 visualizes an implicit clustering into clusters of (start,goal) combinations and shows the results of the clusters of ground truth trajectories (a) and their corresponding implicit clusters (b). In this experiment, start and goal positions are considered factors of variation that are varied and the motion planner learns to generate samples that approximate the original dataset. We control the generation by varying the respective factors of variation and generate paths according to Equation 3. The atomic partitions in this context correspond to pairs of (start,goal) partitions in an euclidean space. In comparison to the ground truth clusters, variations of the paths are generated that mimic the ground truth distribution of expert demonstrations. In the same context, completely new scenarios can be synthesized by algorithms that would allow the model to learn react to the environment, such as GAIL Ho & Ermon (2016).

**Analytical Factors of Variation — Compositional Image Synthesis.** Similar to relaxing start and goal positions for motion planning as factors of variation, placement of objects in an image is also a straightforward relaxation in the euclidean space. In this experiment, we show an application of our approach to compositional generation of images by relaxing several factors of variation required for object composition. More specifically, we relax the placement (x,y) placement coordinates and the number of objects, which gives three analytical factors of variation.

Figure 8a and Figure 8b show exemplary implicit clusters on the CLEVR Relations dataset. We can control the generation by varying the respective factors of variation and generating images according to Equation 3. We emphasize that the underlying model fails to generate coherent clusters near the distribution boundaries, highlight the need for better OOD generalization even in "simple" placement tasks. However, we want to highlight that this kind of task can as well be expressed under Equation 3 as an *Implicit Neural Clustering*.

## D ADDITIONAL COMMENTS AND EXPERIMENTS ON COMBINATORIAL GENERALIZATION

In the relaxed setting with $h^*$, an application to diffusion models with known factors of variation that shows combinatorial generalization under Equation 3 can be given based on the work by Okawa

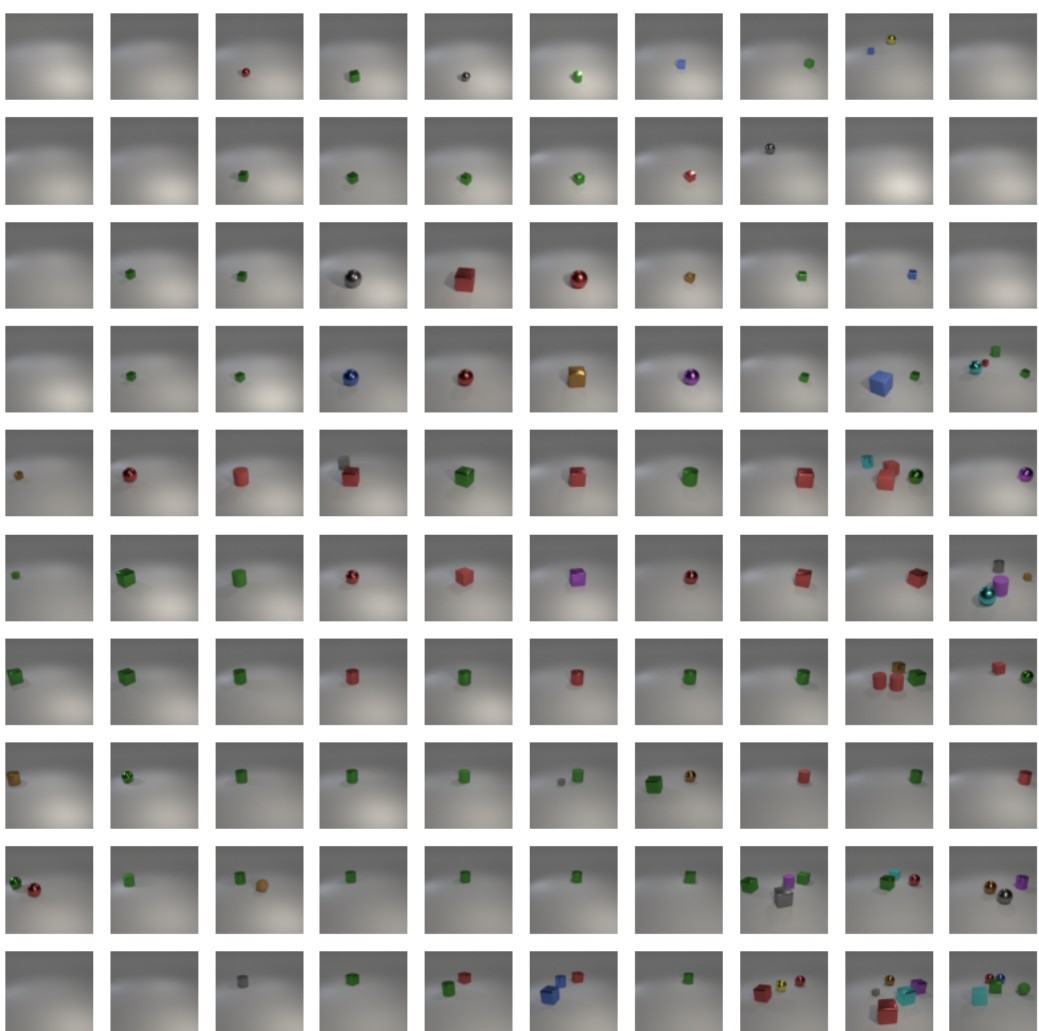

Figure 9: Partitions on x and y with all cross combinations.

et al. (2024). In this experiment, the factors of variation can be naturally partitioned into atomic partitions due to their discrete nature. We reproduce their experiments on combinatorial generalization by training a conditional diffusion model to generate images conditioned on restricted factors of variation. We show the explicit clusters compared to the implicit clusters that can be generated in Figure 10. We can see that the model learns combinatorial generalization to generate small spheres, big blue rectangles, and small red and blue rectangles. Note how this experiment mimics our conceptual illustration of *Implicit Neural Clustering* in Figure 2, showing strong empirical evidence for the correctness of our definition. Finally, we want to highlight that Okawa et al. have rigorously tested when diffusion models achieved combinatorial generalization in this synthetic dataset Okawa et al. (2024). Their overall experiments provide empirical evidence for the validity and practicality of our approach while satisfying all of our conditions ($A.H$, $A.G1$, $A.G2$, $A.G3$) in the special case of relaxation to $h^*$.

# E   ADDITIONAL RESULTS FOR DSPRITES

We provide the full implicit multi-partition clustering of dSprites.

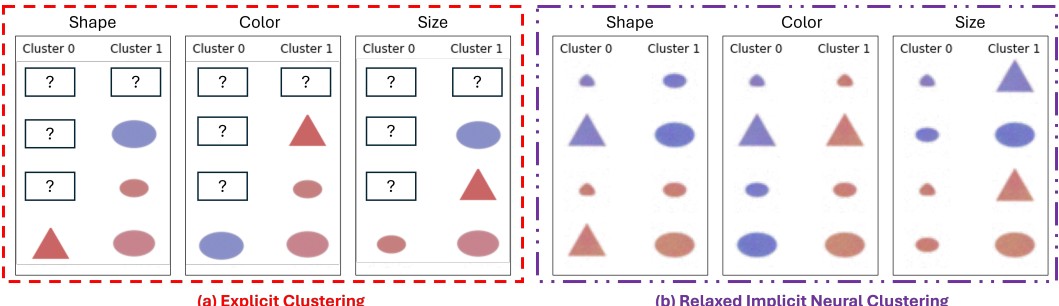

Figure 10: Relaxed implicit neural clustering of shapes (right) effectively approximates and generalizes to novel cross combinations of the underlying ground truth distribution (left). This experiment provides empirical evidence that diffusion models trained on synthetic data can satisfy the most difficult part of moving beyond the data distribution with *Implicit Neural Clustering*. In this example, the generative model generalizes to being able to synthesize elements not seen in the training dataset, hence filling in the gaps in the data distribution as previously illustrated in Figure 2.

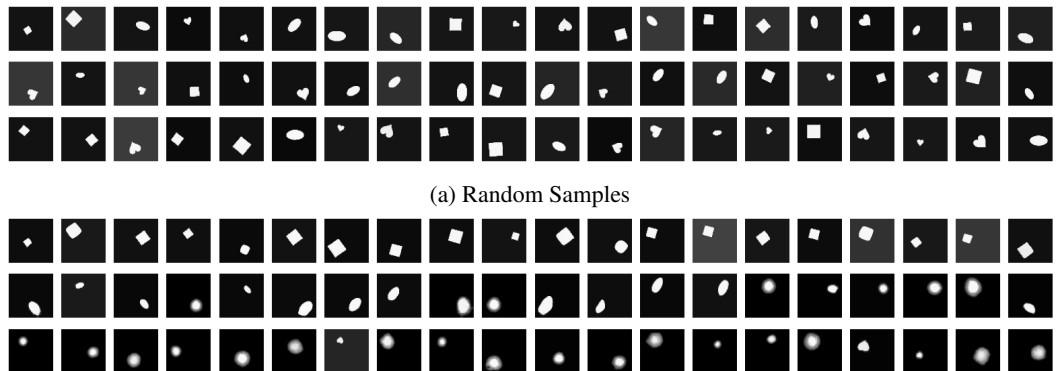

Figure 11: dSprites: Implicit clusters for object shape

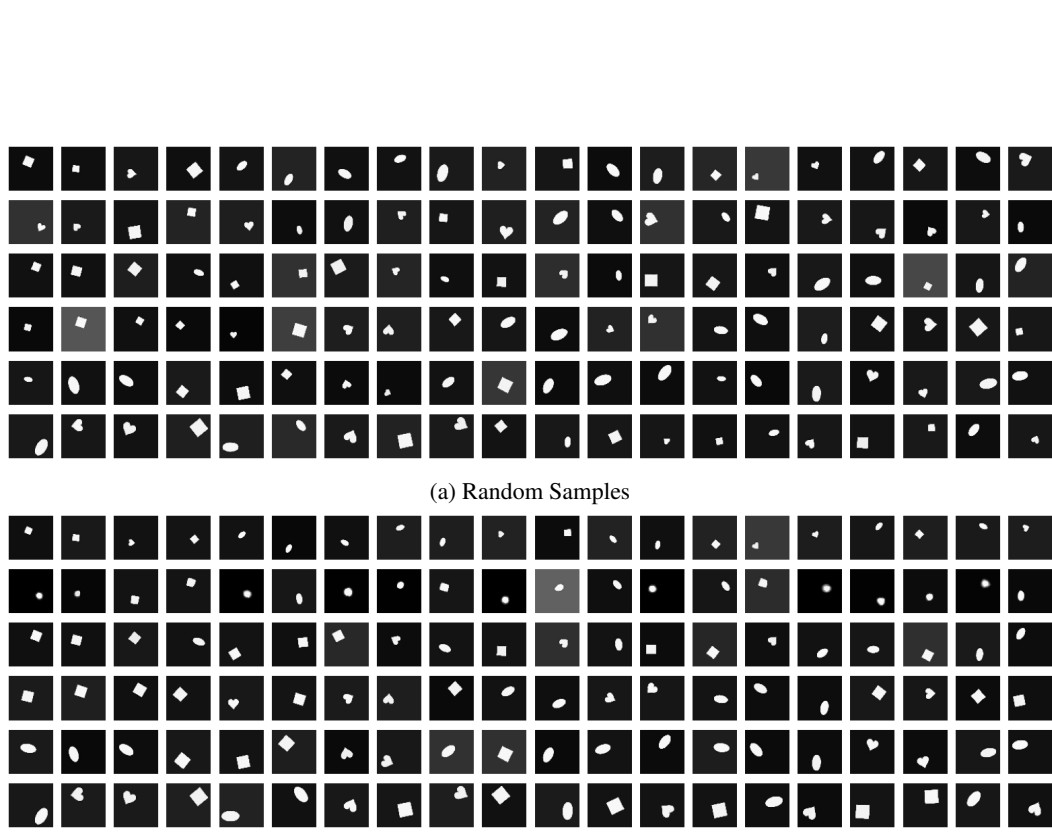

(a) Random Samples

(b) w/ Atomic Group Actions

Figure 12: dSprites: Implicit clusters for object size

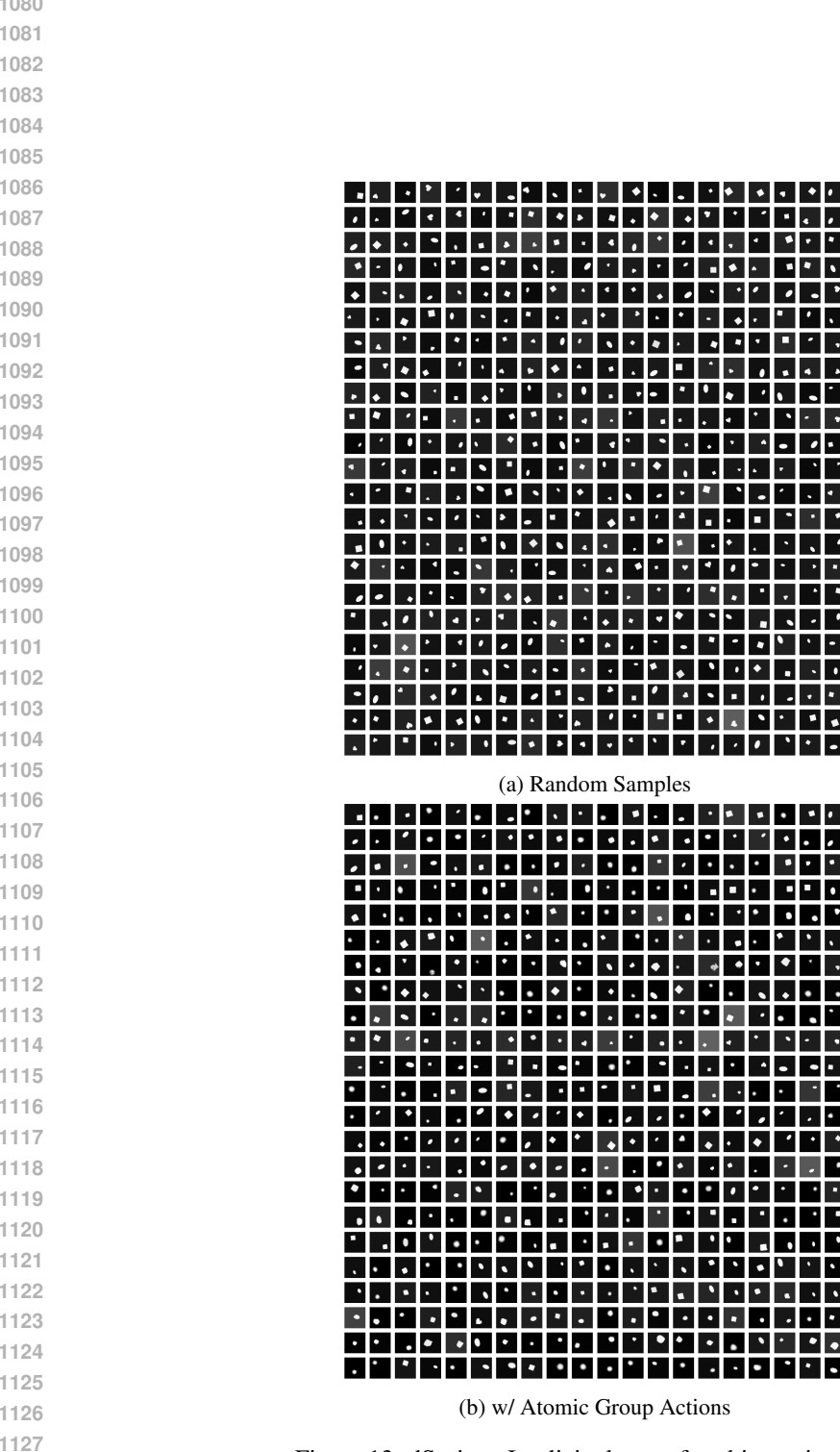

(a) Random Samples

(b) w/ Atomic Group Actions

Figure 13: dSprites: Implicit clusters for object orientation

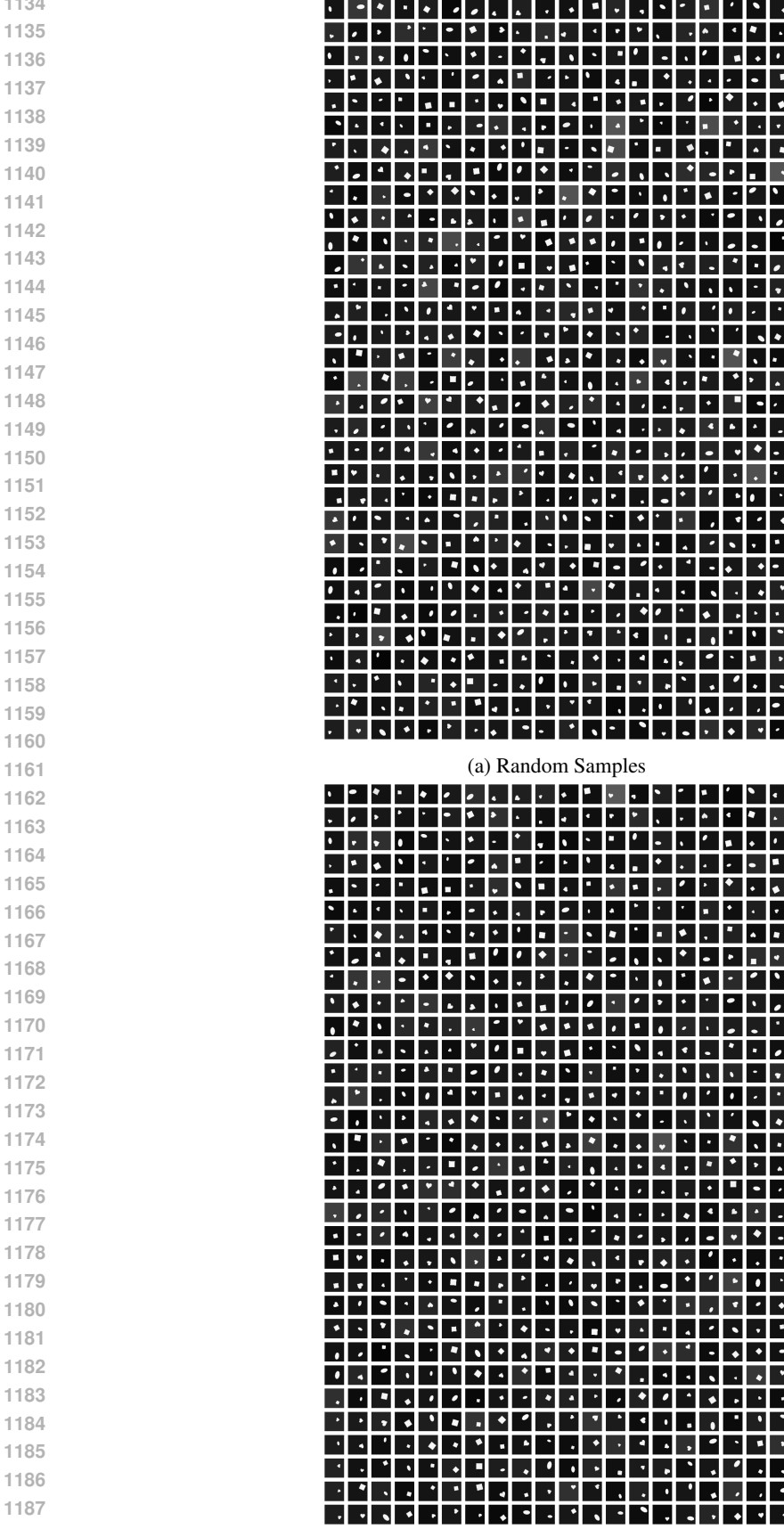

(a) Random Samples

(b) w/ Atomic Group Actions

Figure 14: dSprites: Implicit Clusters for object X coordinate

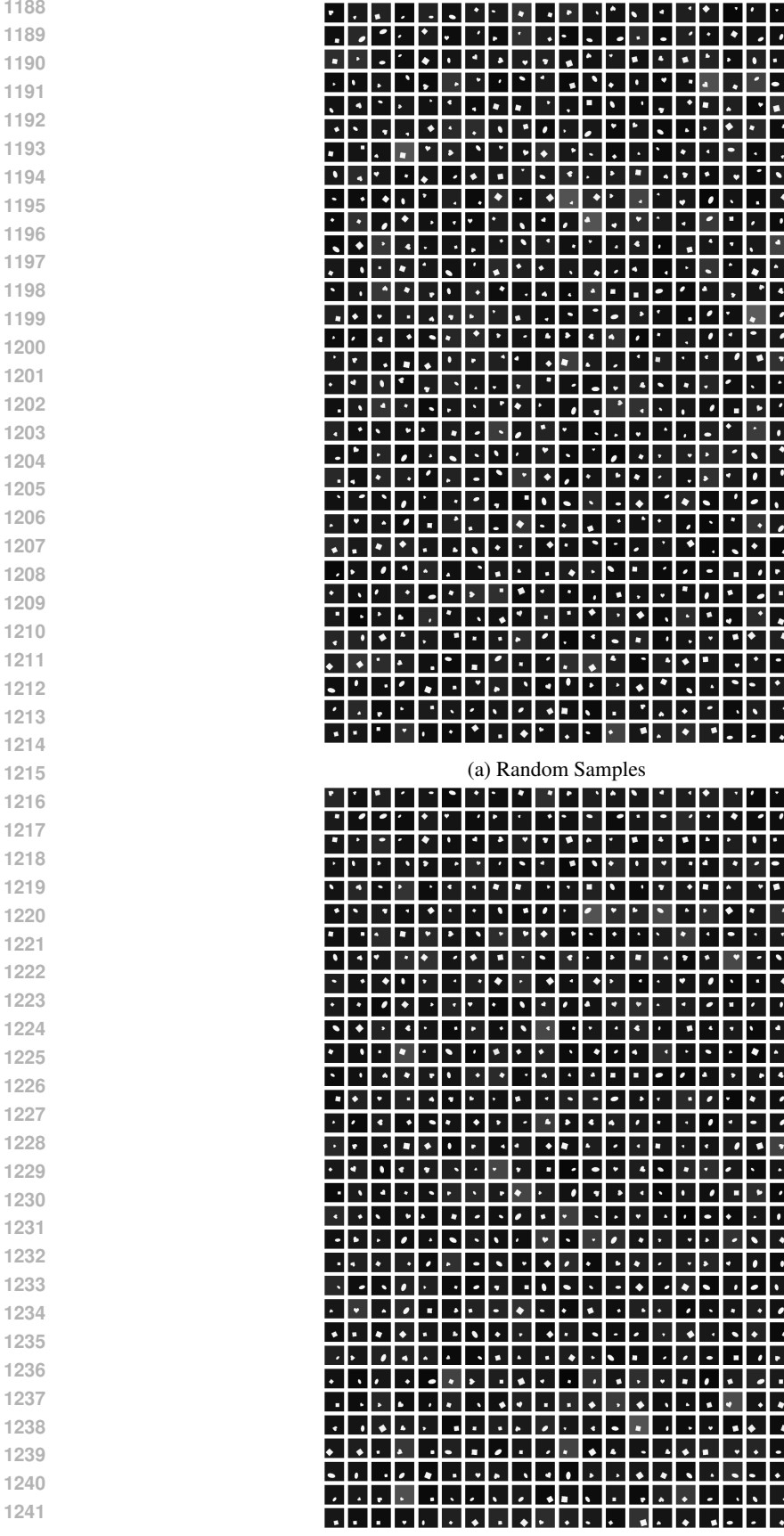

(a) Random Samples

(b) w/ Atomic Group Actions

Figure 15: dSprites: Implicit Clusters for object Y coordinate

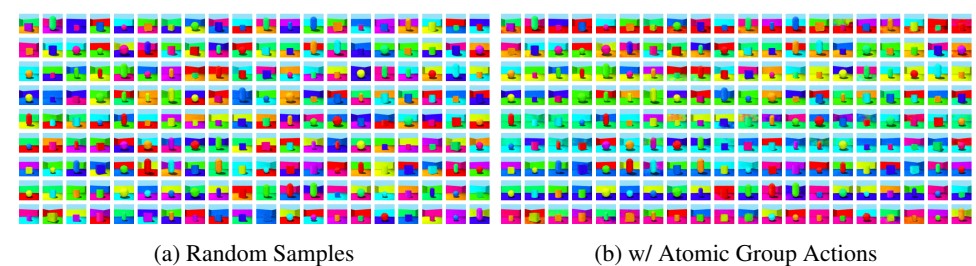

(a) Random Samples                        (b) w/ Atomic Group Actions

Figure 16: 3DShapes: Implicit clusters for floor color

# F    ADDITIONAL RESULTS FOR 3DSHAPES

We provide the full implicit multi-partition clustering of 3DShapes.

## F.1    IMPLICIT NEURAL CLUSTERINGS

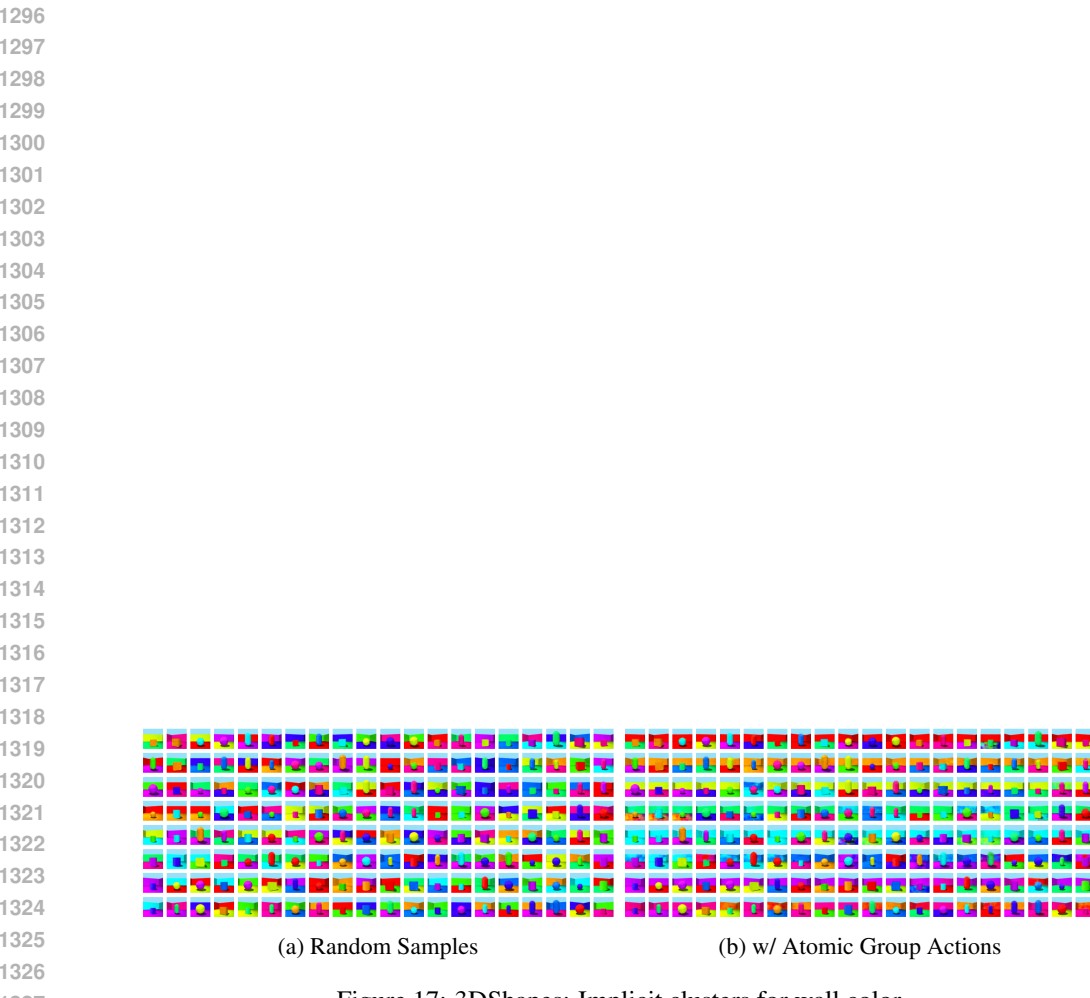

(a) Random Samples                (b) w/ Atomic Group Actions

Figure 17: 3DShapes: Implicit clusters for wall color

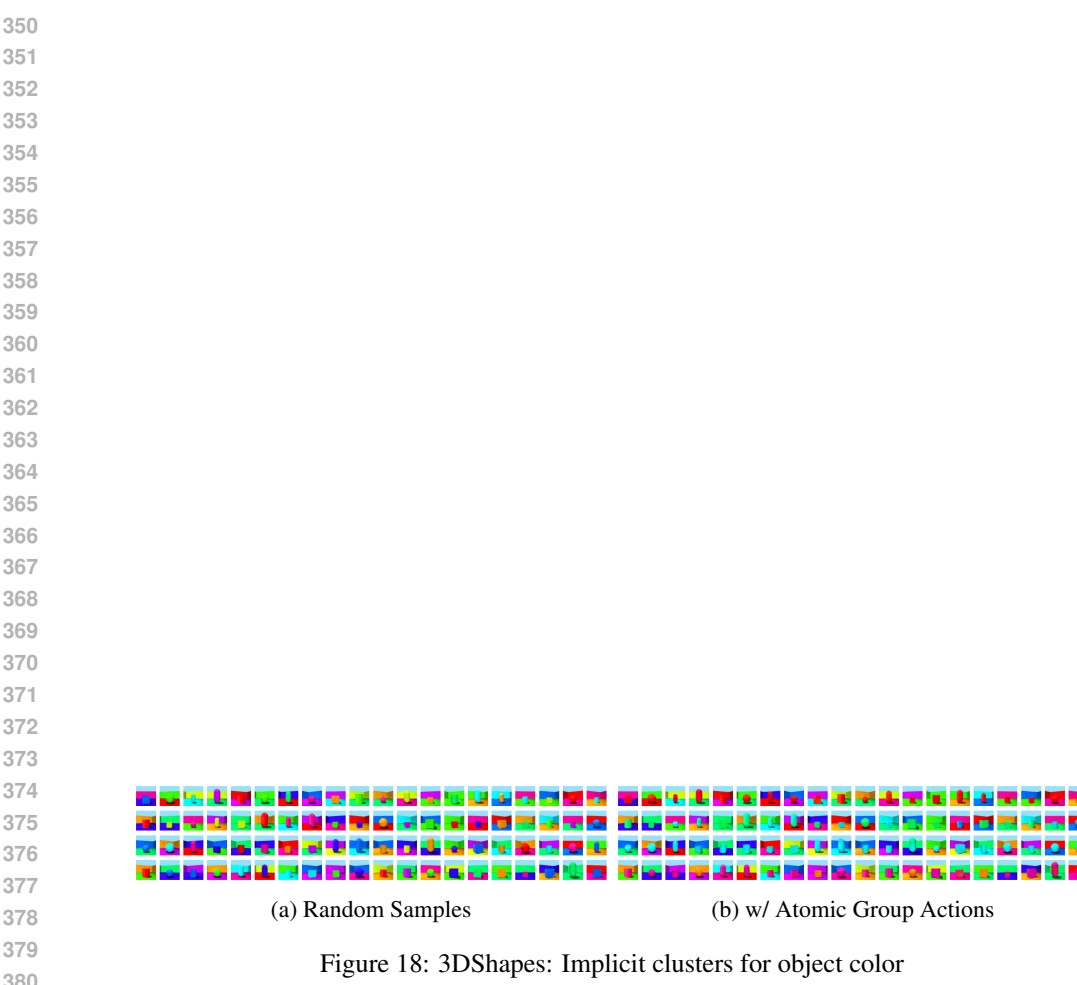

(a) Random Samples        (b) w/ Atomic Group Actions

Figure 18: 3DShapes: Implicit clusters for object color

1404
1405
1406
1407
1408
1409
1410
1411
1412
1413
1414
1415
1416
1417
1418
1419
1420
1421
1422
1423
1424
1425
1426
1427
1428
1429
1430
1431
1432
1433
1434
1435
1436
1437
1438
1439
1440
1441
1442
1443
1444
1445
1446
1447
1448
1449
1450
1451
1452
1453
1454
1455
1456
1457

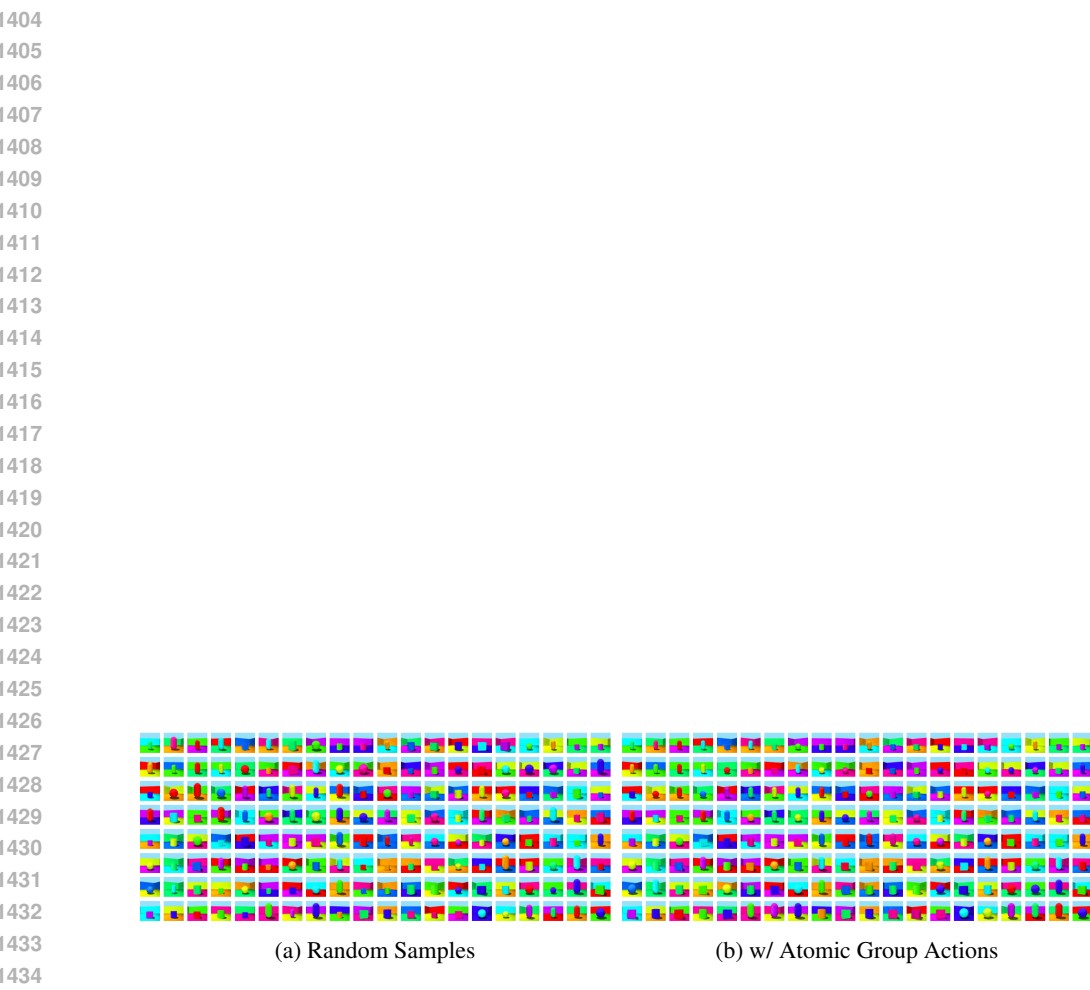

(a) Random Samples                     (b) w/ Atomic Group Actions

Figure 19: 3DShapes: Implicit clusters for object size

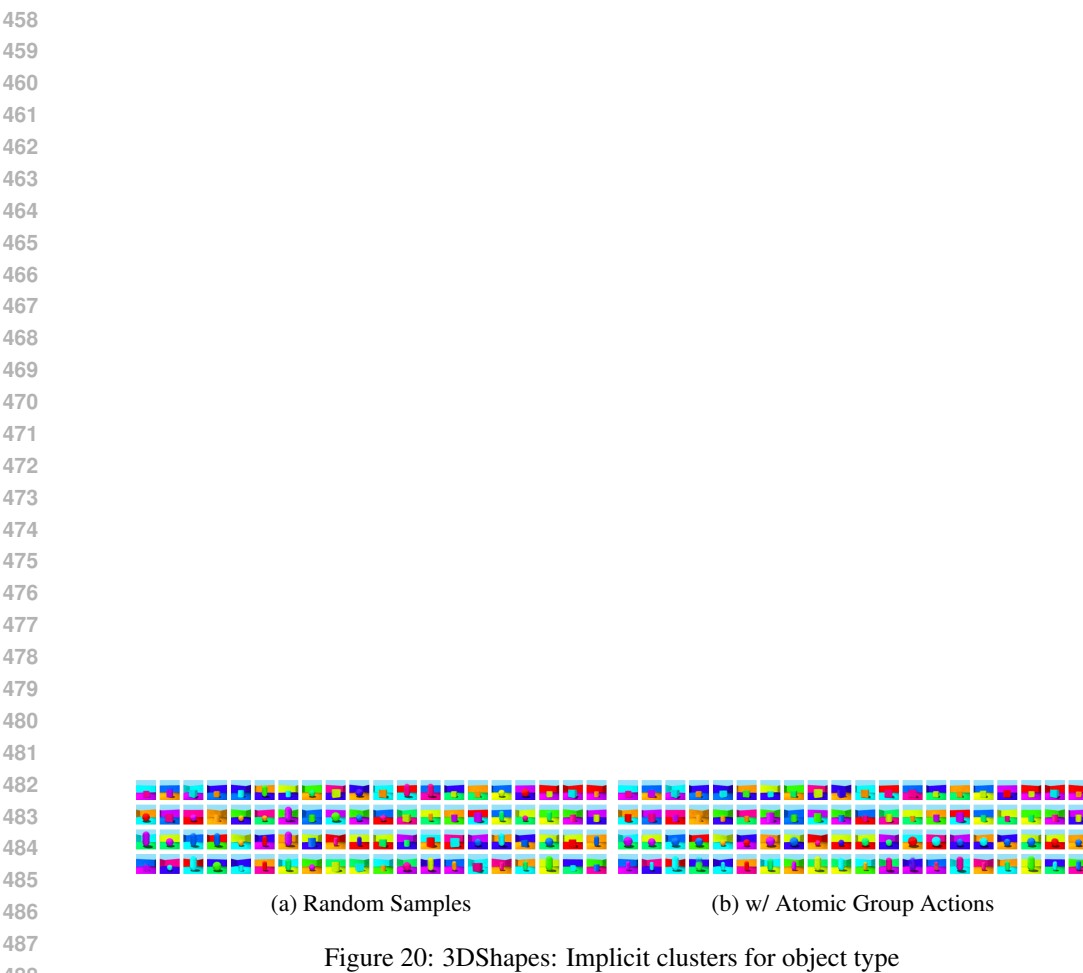

(a) Random Samples          (b) w/ Atomic Group Actions

Figure 20: 3DShapes: Implicit clusters for object type

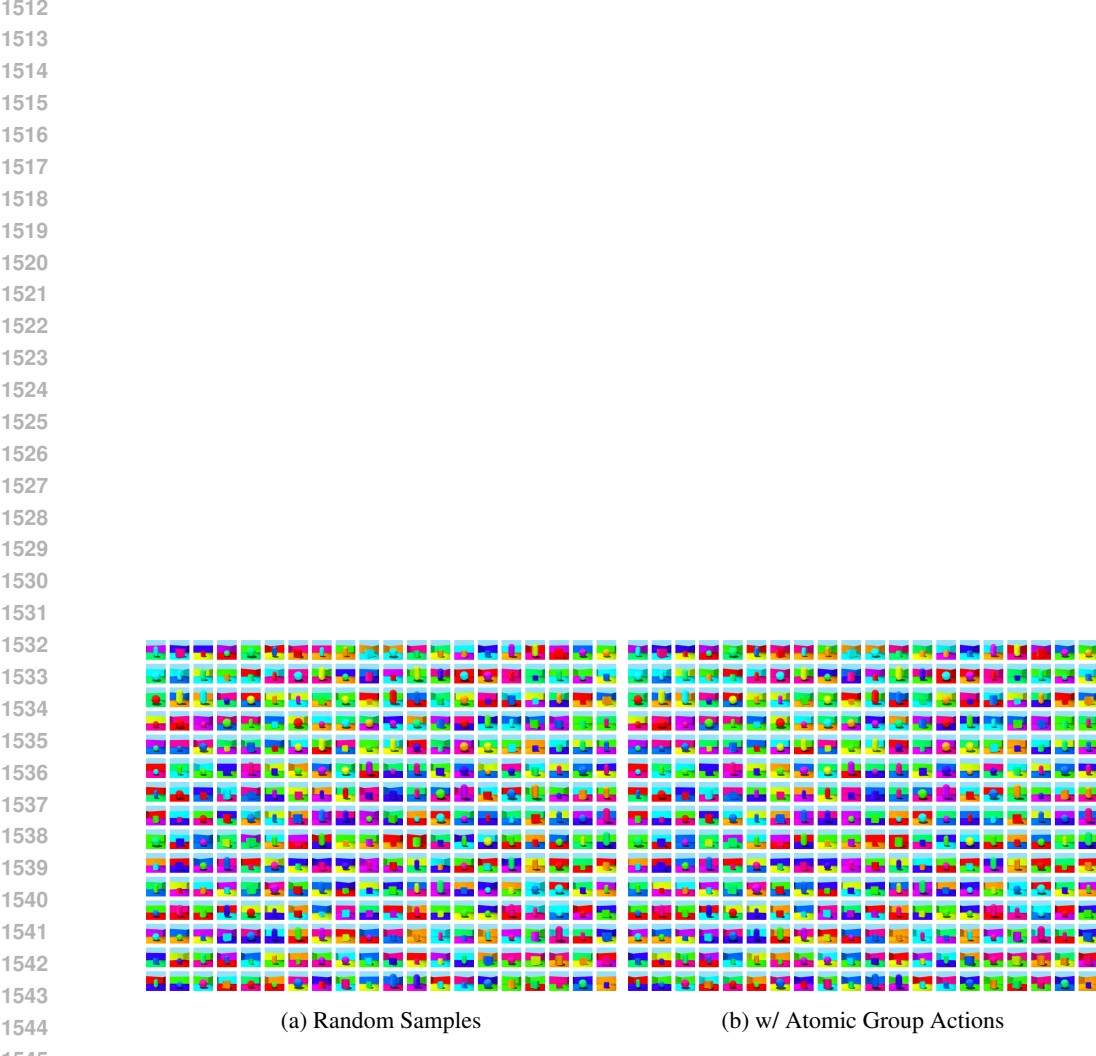

(a) Random Samples        (b) w/ Atomic Group Actions

Figure 21: 3DShapes: Implicit clusters for camera angle

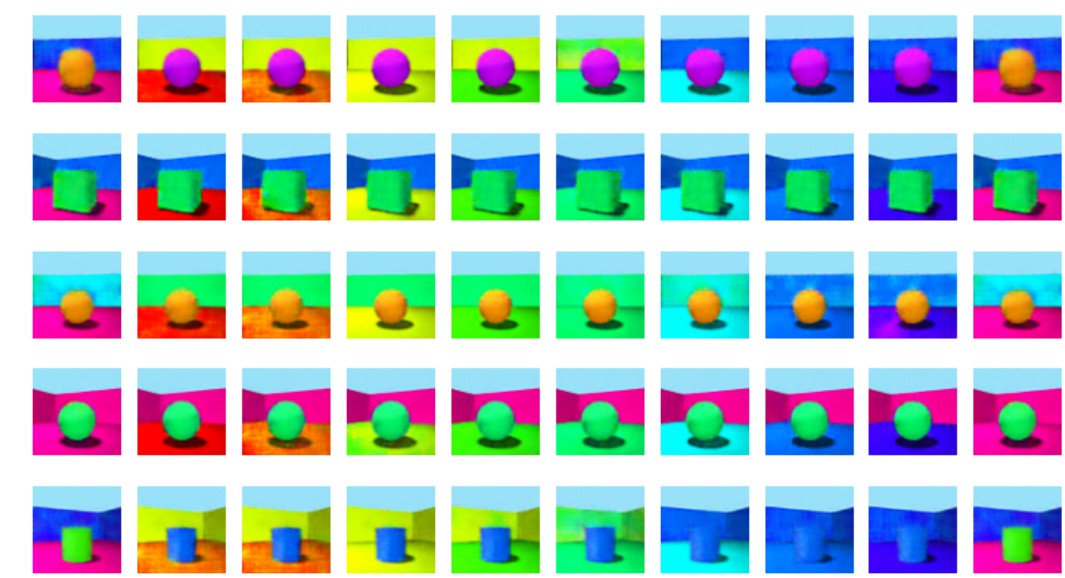

Figure 22: 3DShapes: Latent traversal with atomic group actions, floor color

## F.2 LATENT TRAVERSALS WITH ATOMIC GROUP ACTIONS

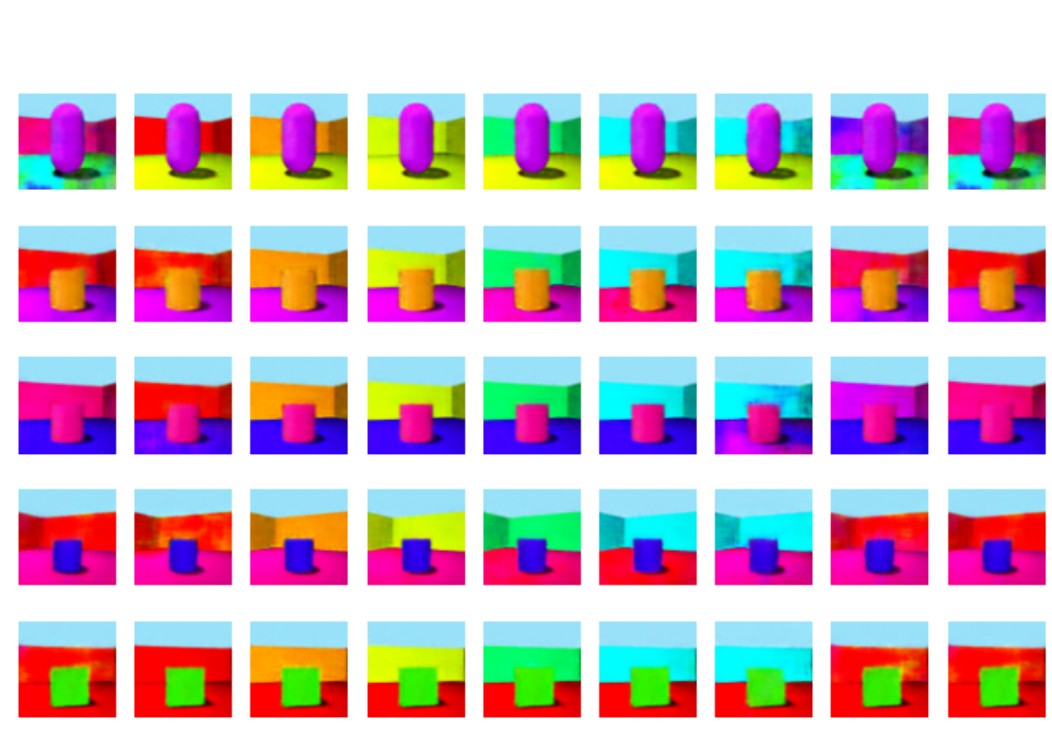

Figure 23: 3DShapes: Latent traversal with atomic group actions, wall color

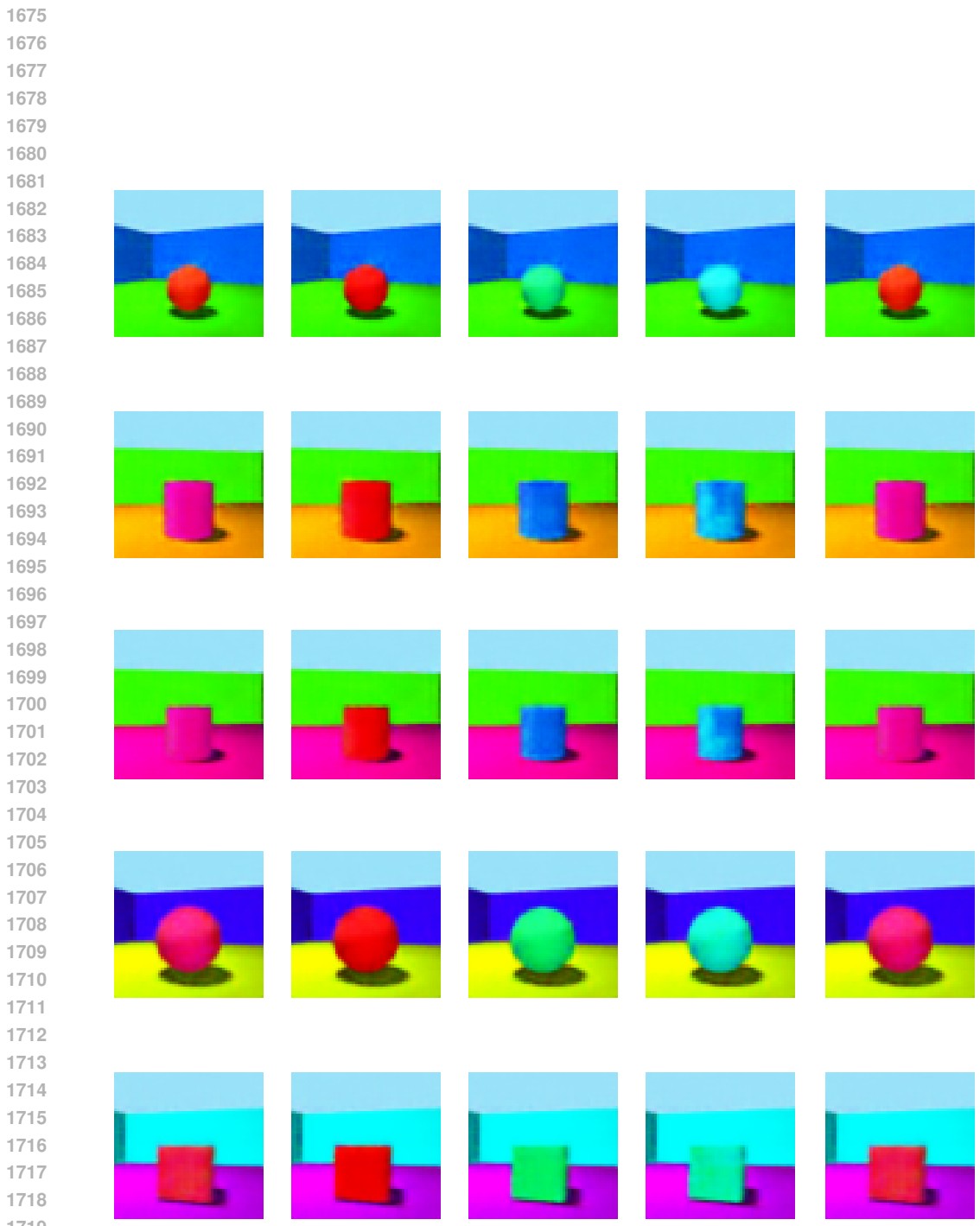

Figure 24: 3DShapes: Latent traversal with atomic group actions, object color

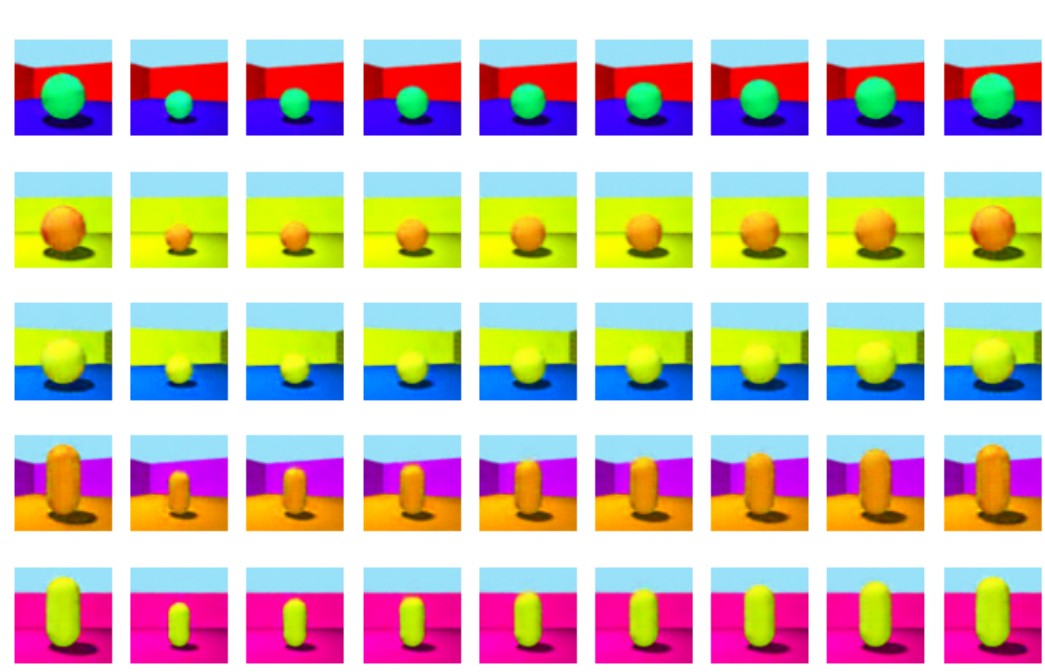

Figure 25: 3DShapes: Latent traversal with atomic group actions, object size

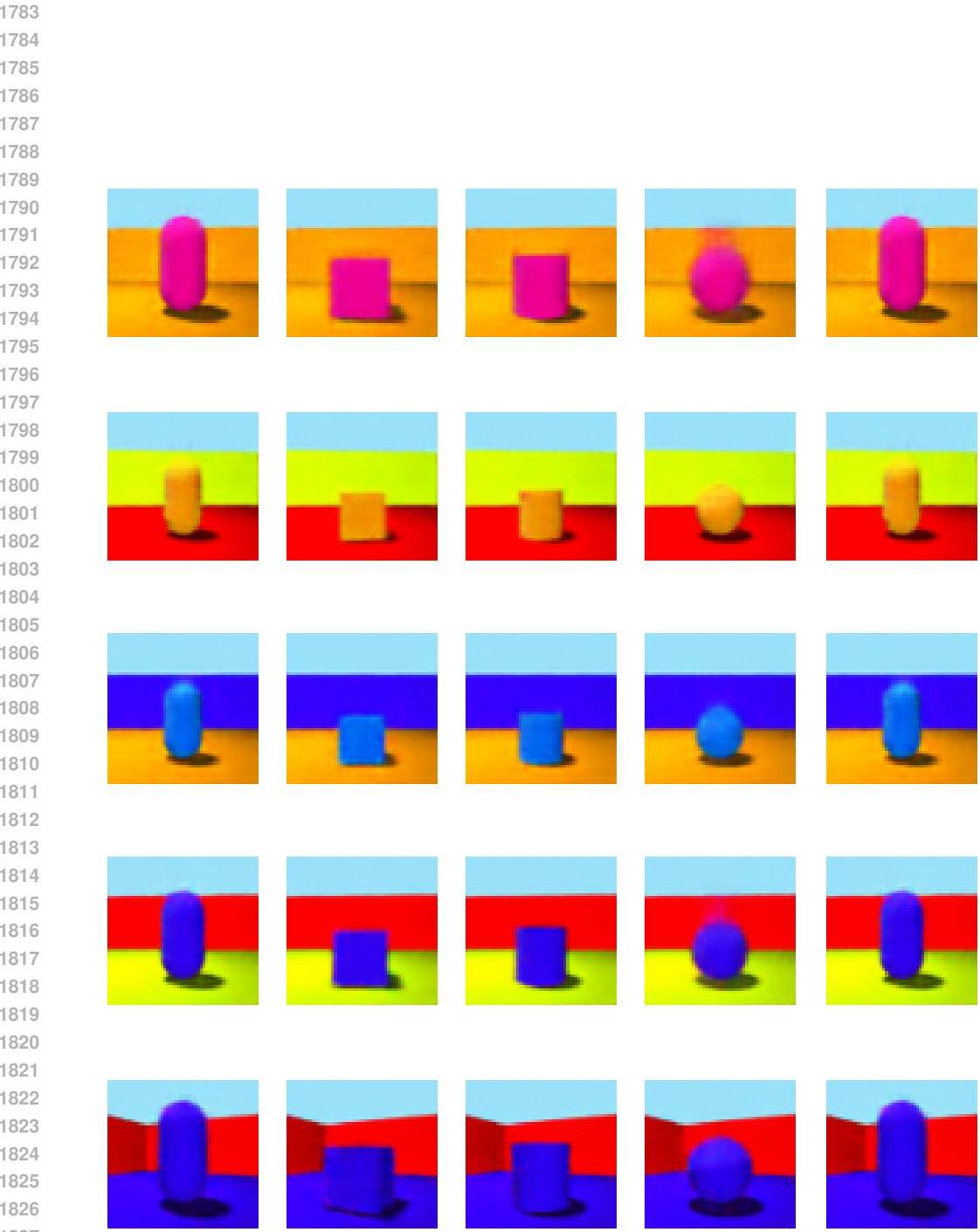

Figure 26: 3DShapes: Latent traversal with atomic group actions, object shape

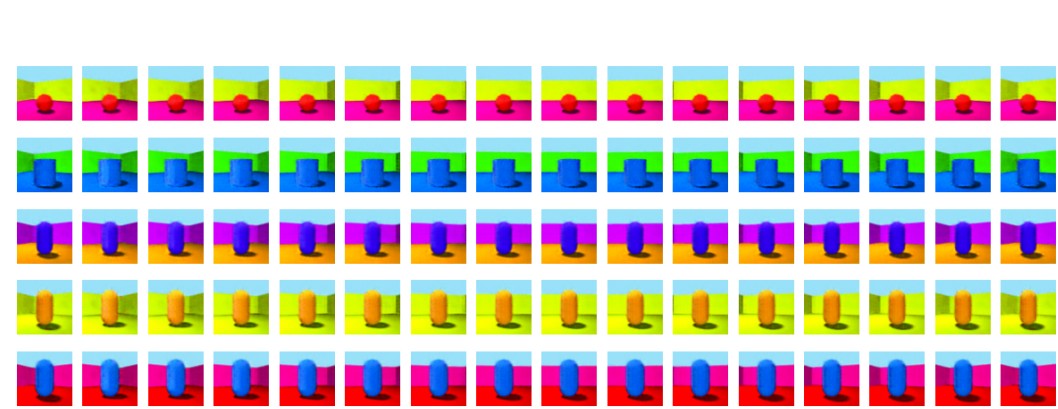

Figure 27: 3DShapes: Latent traversal with atomic group actions, viewing angle

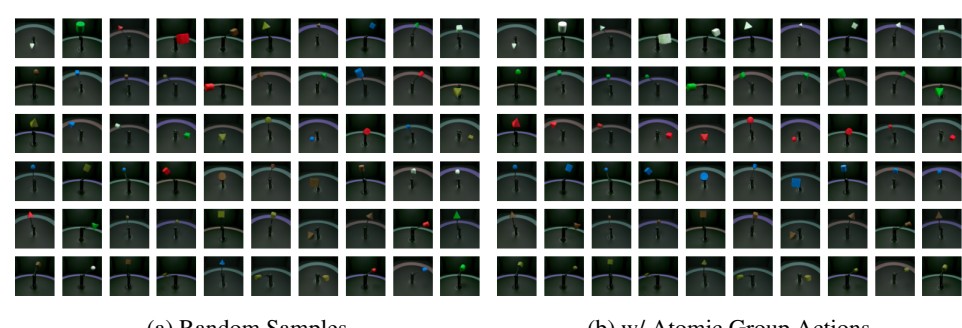

(a) Random Samples        (b) w/ Atomic Group Actions

Figure 28: MPI3D: Implicit clusters for object color

# G   ADDITIONAL RESULTS ON MPI3D

We provide the full implicit multi-partition clustering of MPI3D.

## G.1   IMPLICIT NEURAL CLUSTERINGS

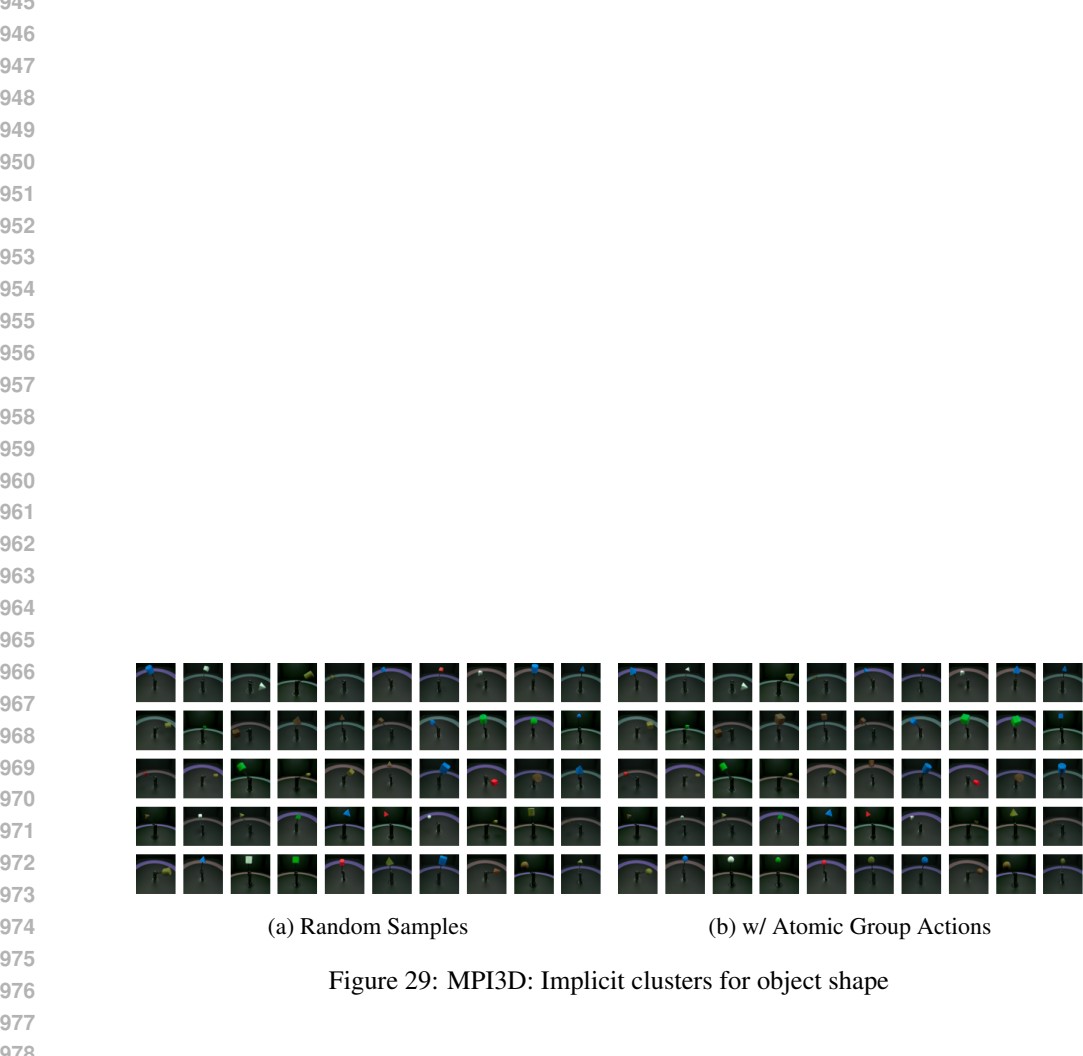

(a) Random Samples                    (b) w/ Atomic Group Actions

Figure 29: MPI3D: Implicit clusters for object shape

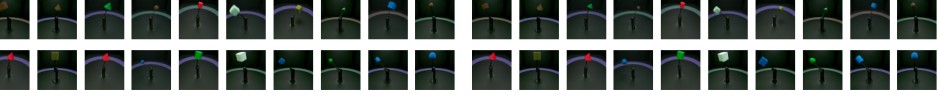

(a) Random Samples                (b) w/ Atomic Group Actions

Figure 30: MPI3D: Implicit clusters for object size

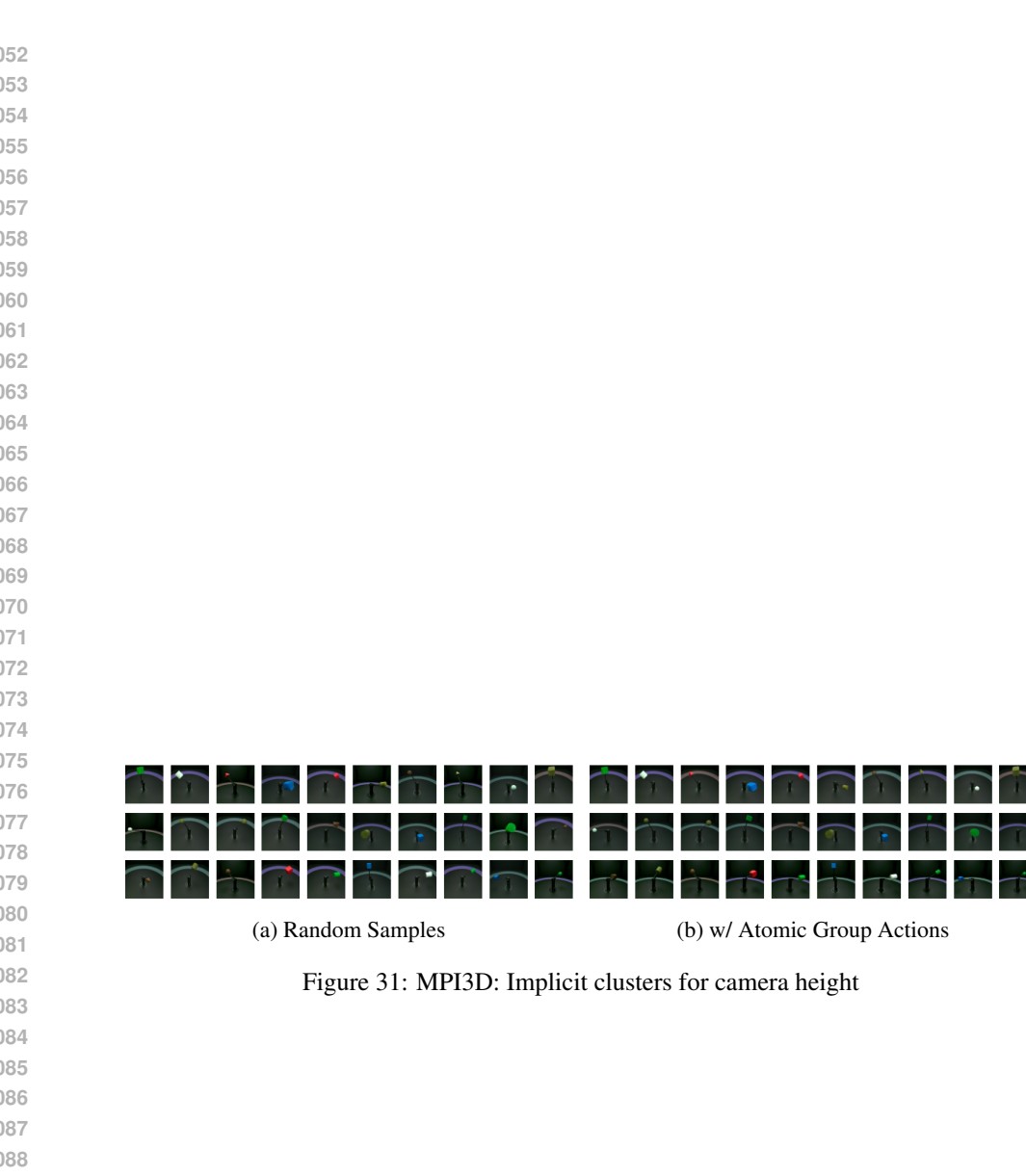

(a) Random Samples                    (b) w/ Atomic Group Actions

Figure 31: MPI3D: Implicit clusters for camera height

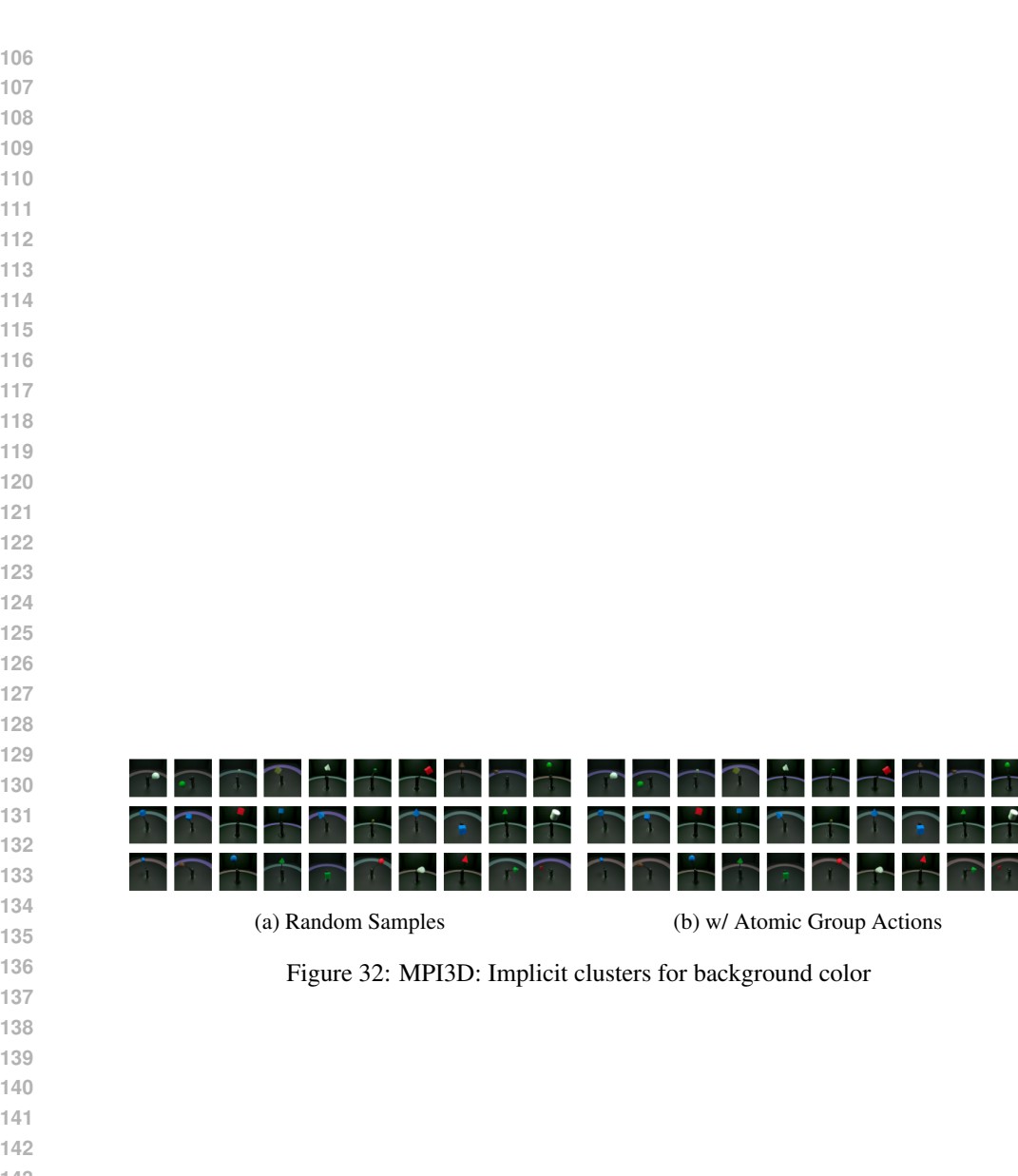

(a) Random Samples        (b) w/ Atomic Group Actions

Figure 32: MPI3D: Implicit clusters for background color

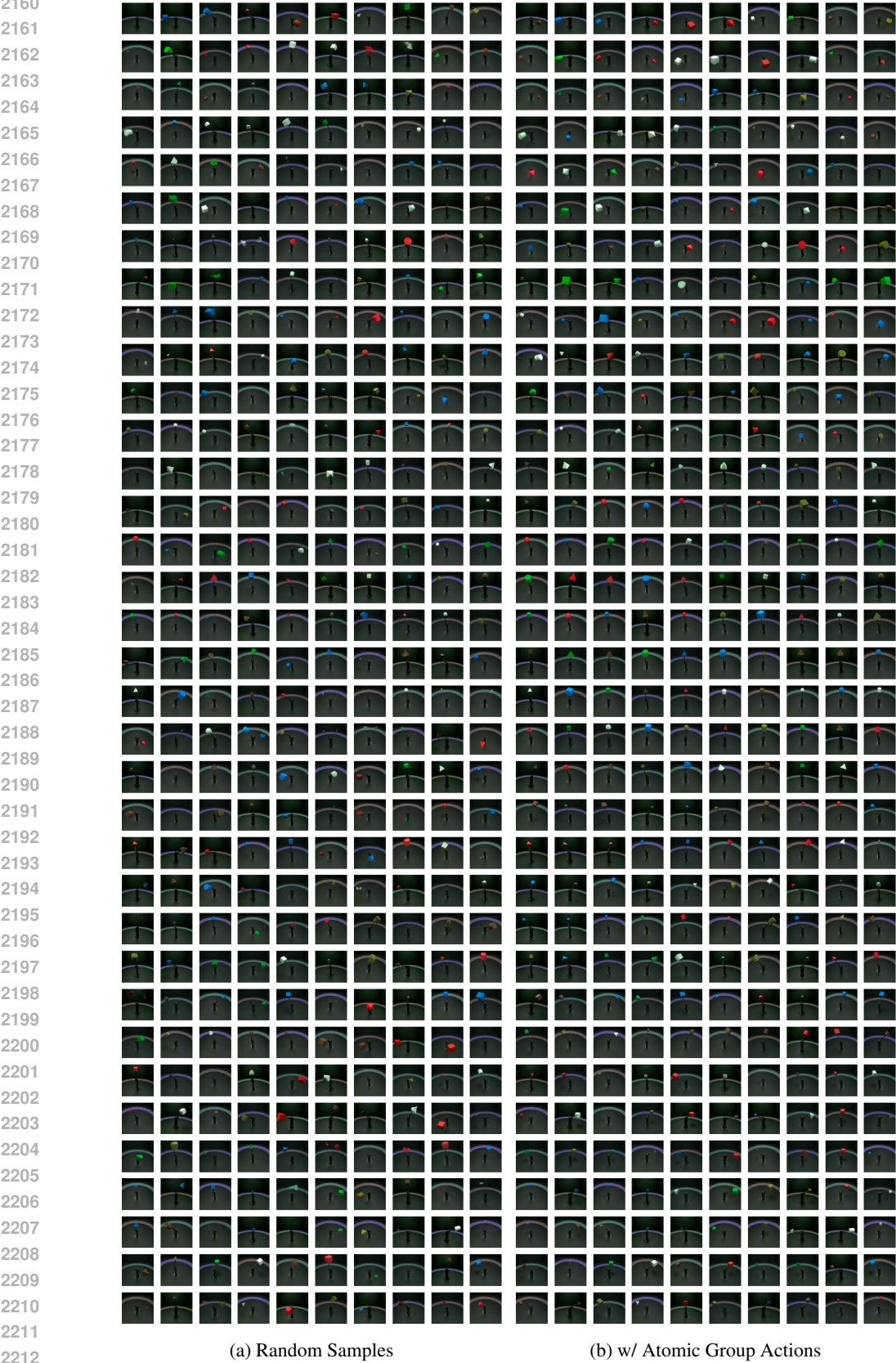

(a) Random Samples         (b) w/ Atomic Group Actions

Figure 33: MPI3D: Implicit clusters for horizontal axis

2214
2215
2216
2217
2218
2219
2220
2221
2222
2223
2224
2225
2226
2227
2228
2229
2230
2231
2232
2233
2234
2235
2236
2237
2238
2239
2240
2241
2242
2243
2244
2245
2246
2247
2248
2249
2250
2251
2252
2253
2254
2255
2256
2257
2258
2259
2260
2261
2262
2263
2264
2265
2266
2267

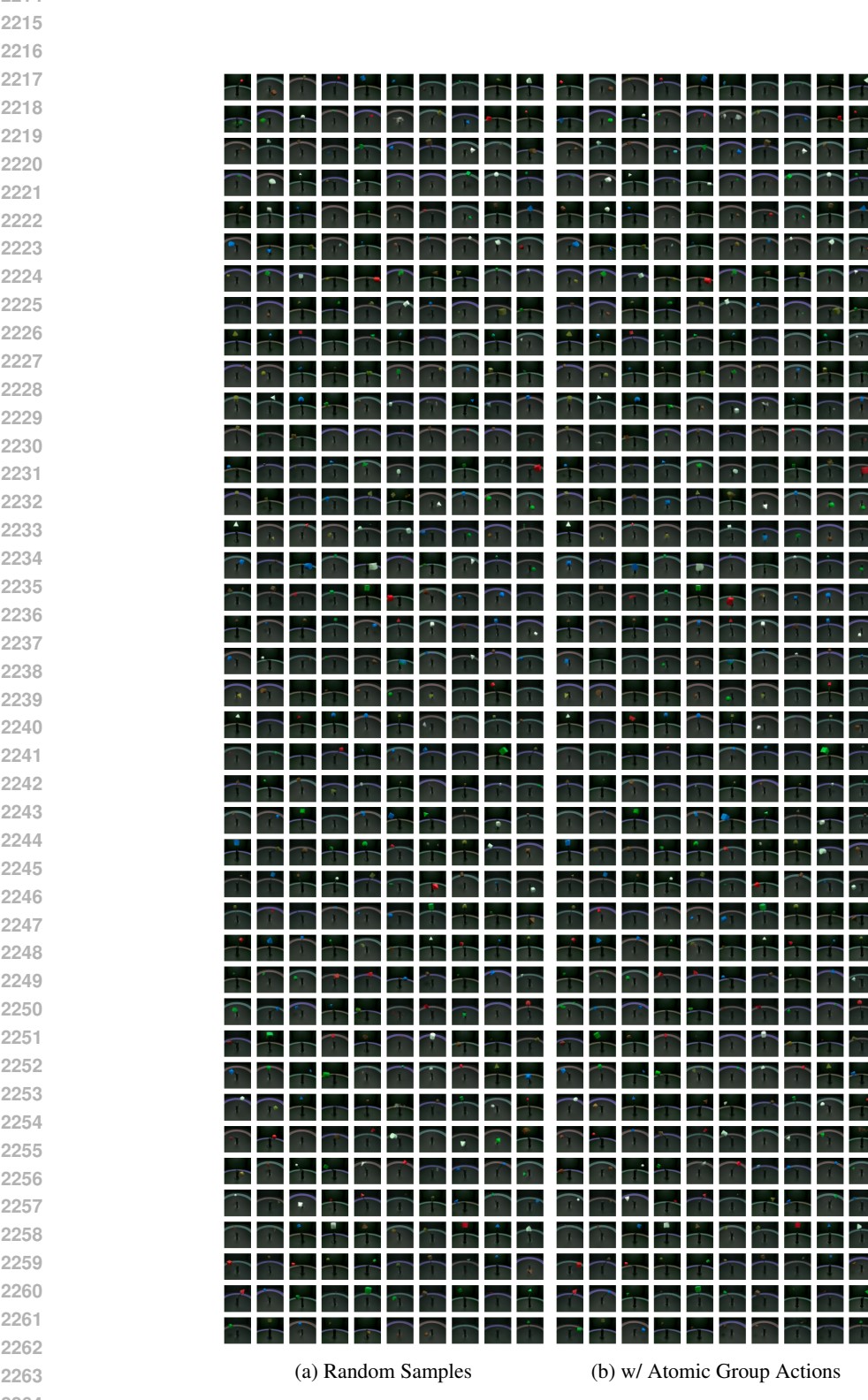

(a) Random Samples     (b) w/ Atomic Group Actions

Figure 34: MPI3D: Implicit clusters for vertical axis

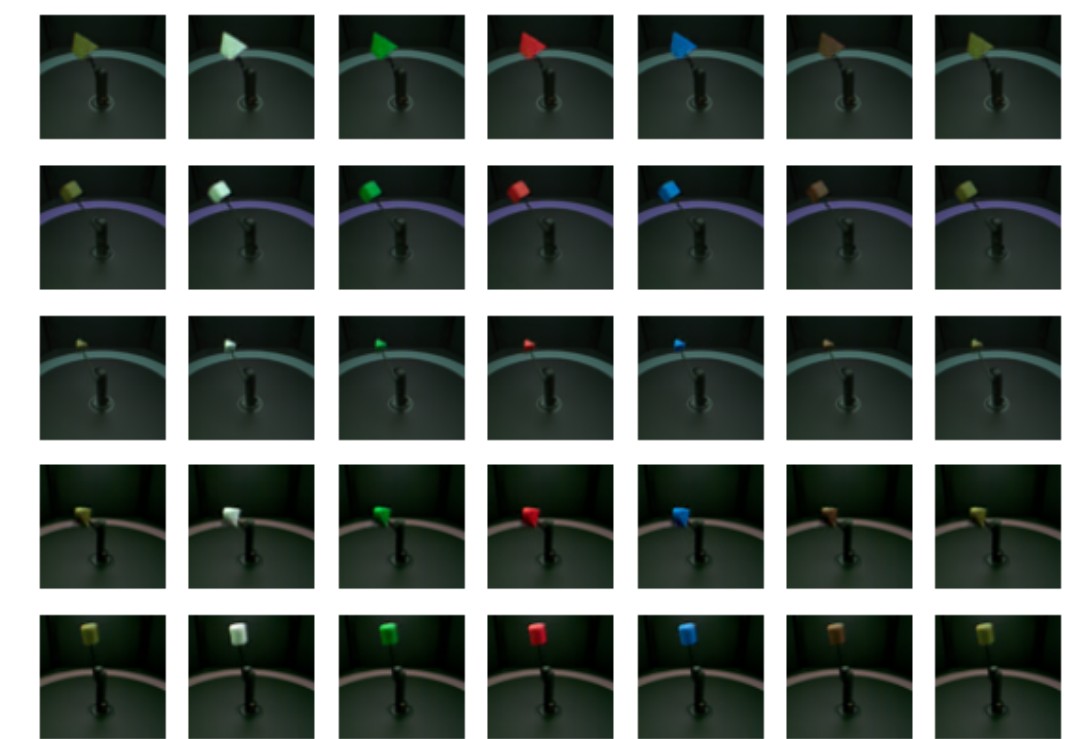

Figure 35: MPI3D: Latent traversal with atomic group actions, object color

## G.2 LATENT TRAVERSALS WITH ATOMIC GROUP ACTIONS

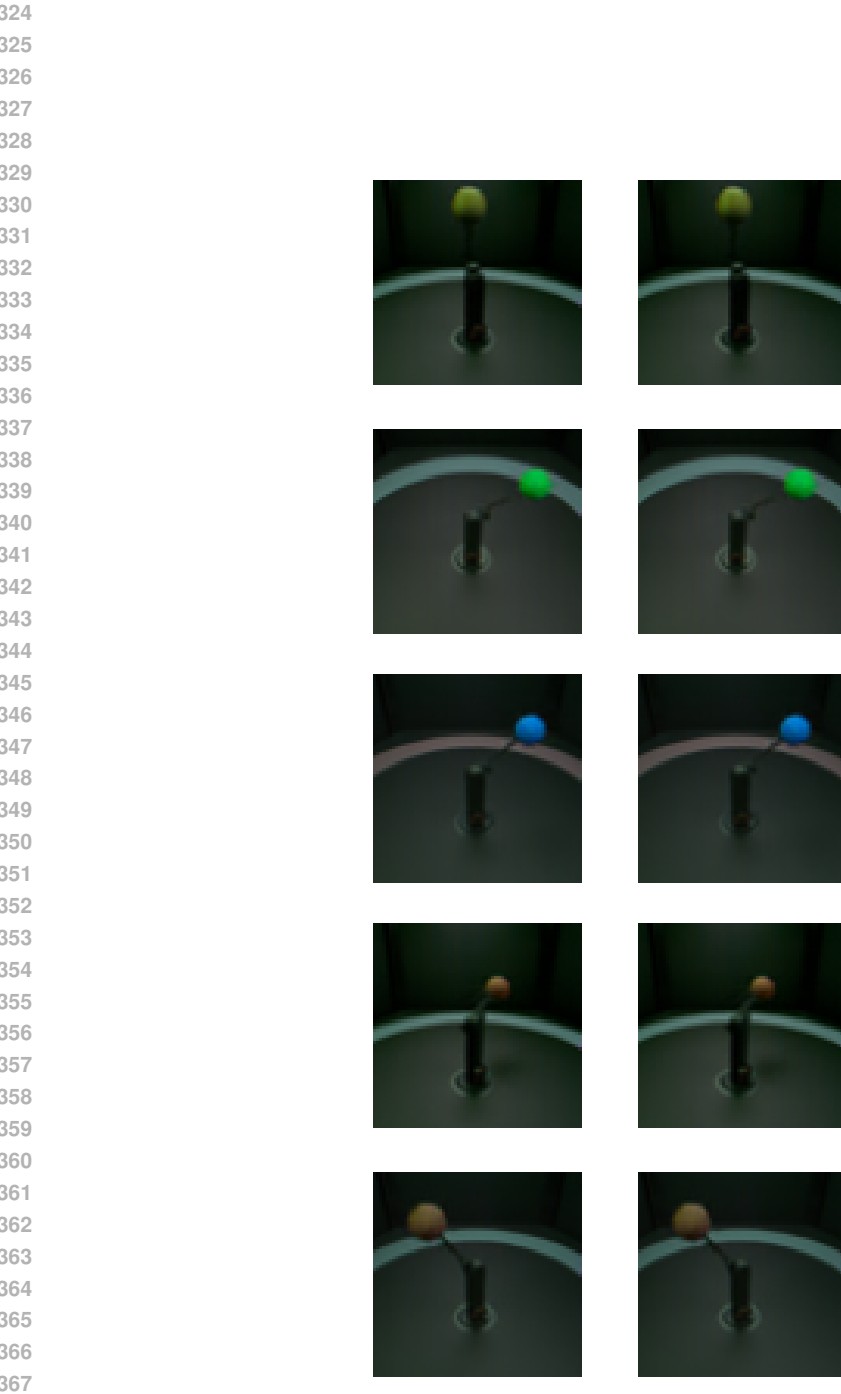

Figure 36: MPI3D: Latent traversal with atomic group actions, object shape

2376
2377
2378
2379
2380
2381
2382
2383
2384
2385
2386
2387
2388
2389
2390
2391
2392
2393
2394
2395
2396
2397
2398
2399
2400
2401
2402
2403
2404
2405
2406
2407
2408
2409
2410
2411
2412
2413
2414
2415
2416
2417
2418
2419
2420
2421
2422
2423
2424
2425
2426
2427
2428
2429

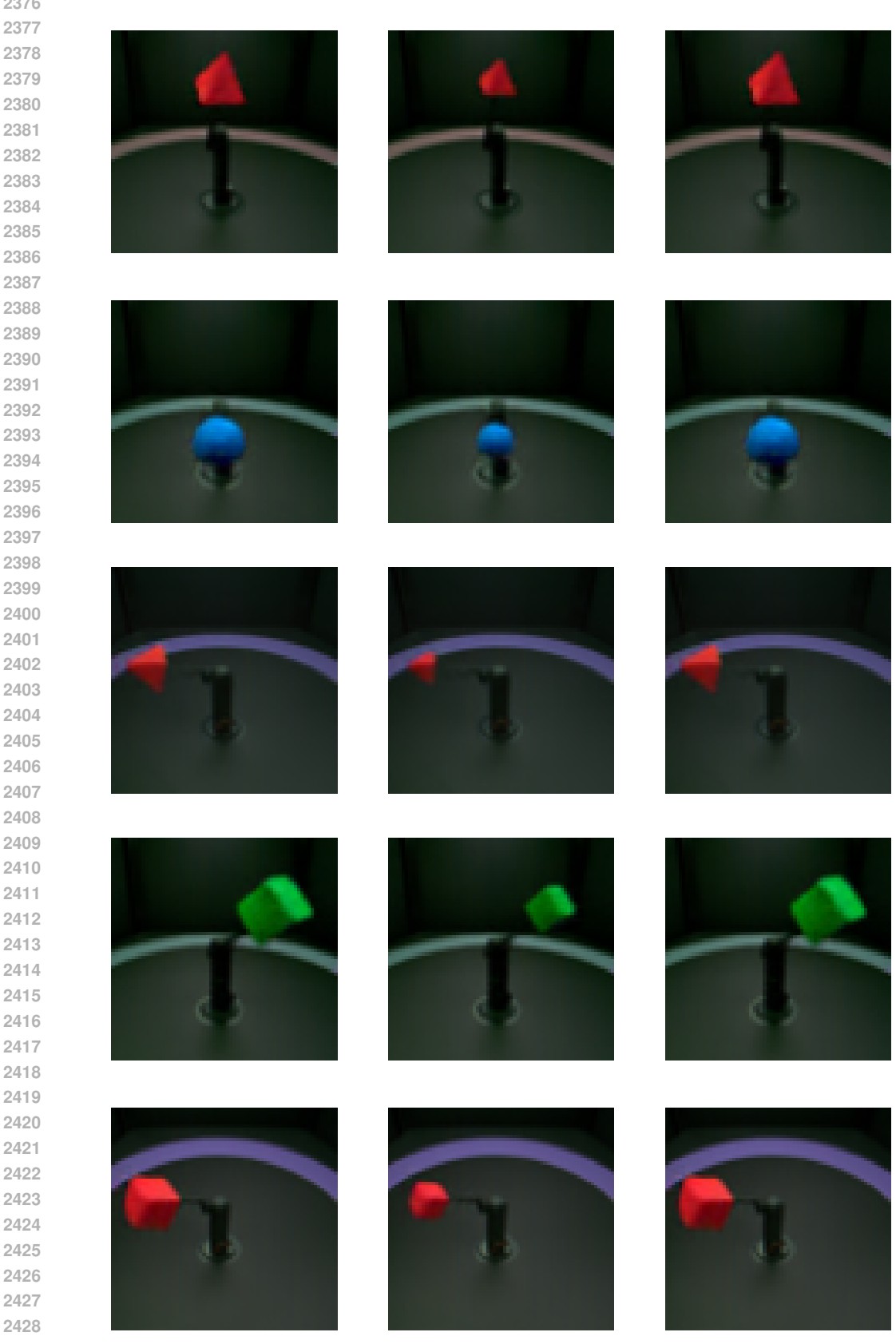

Figure 37: MPI3D: Latent traversal with atomic group actions, object size

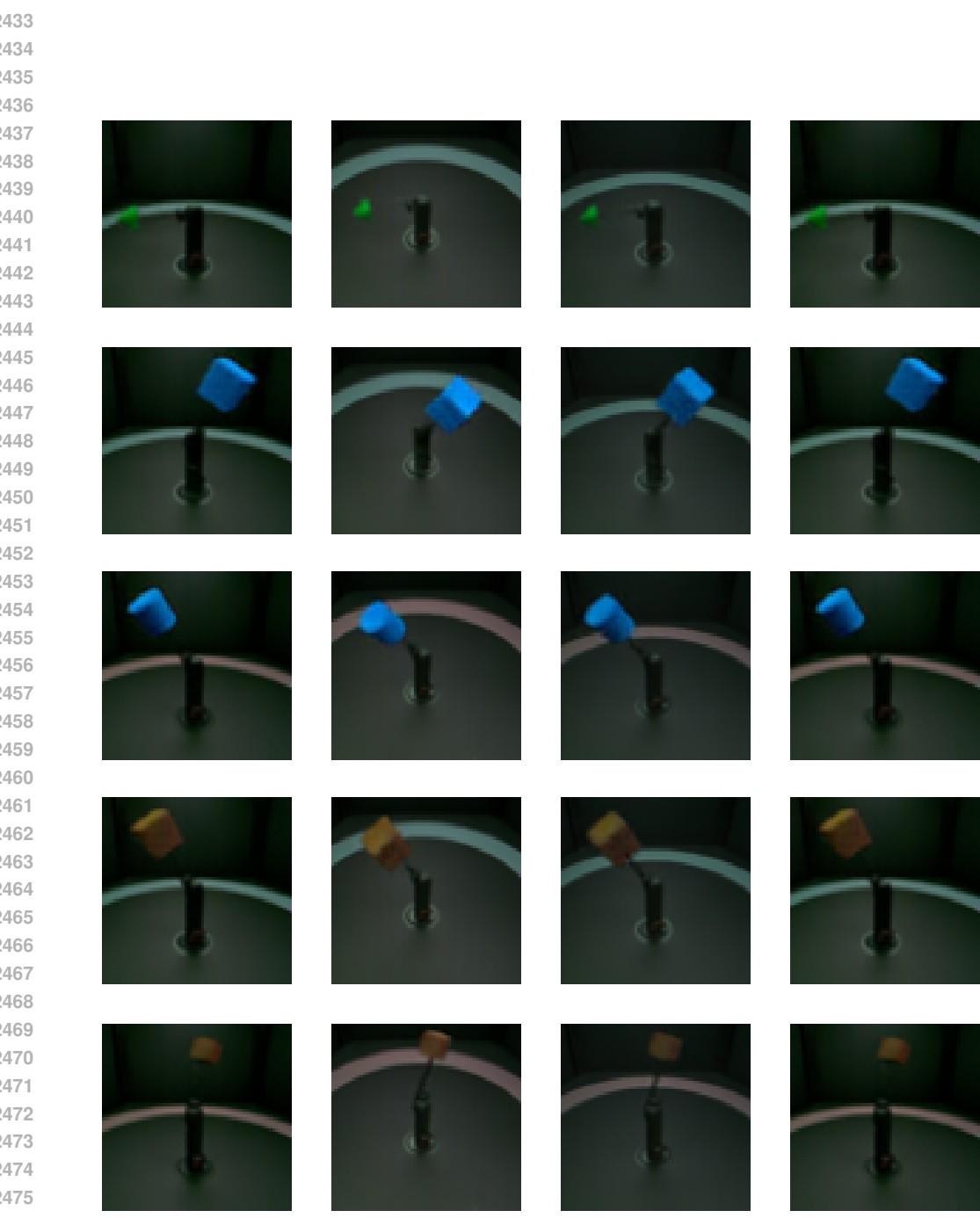

Figure 38: MPI3D: Latent traversal with atomic group actions, camera height

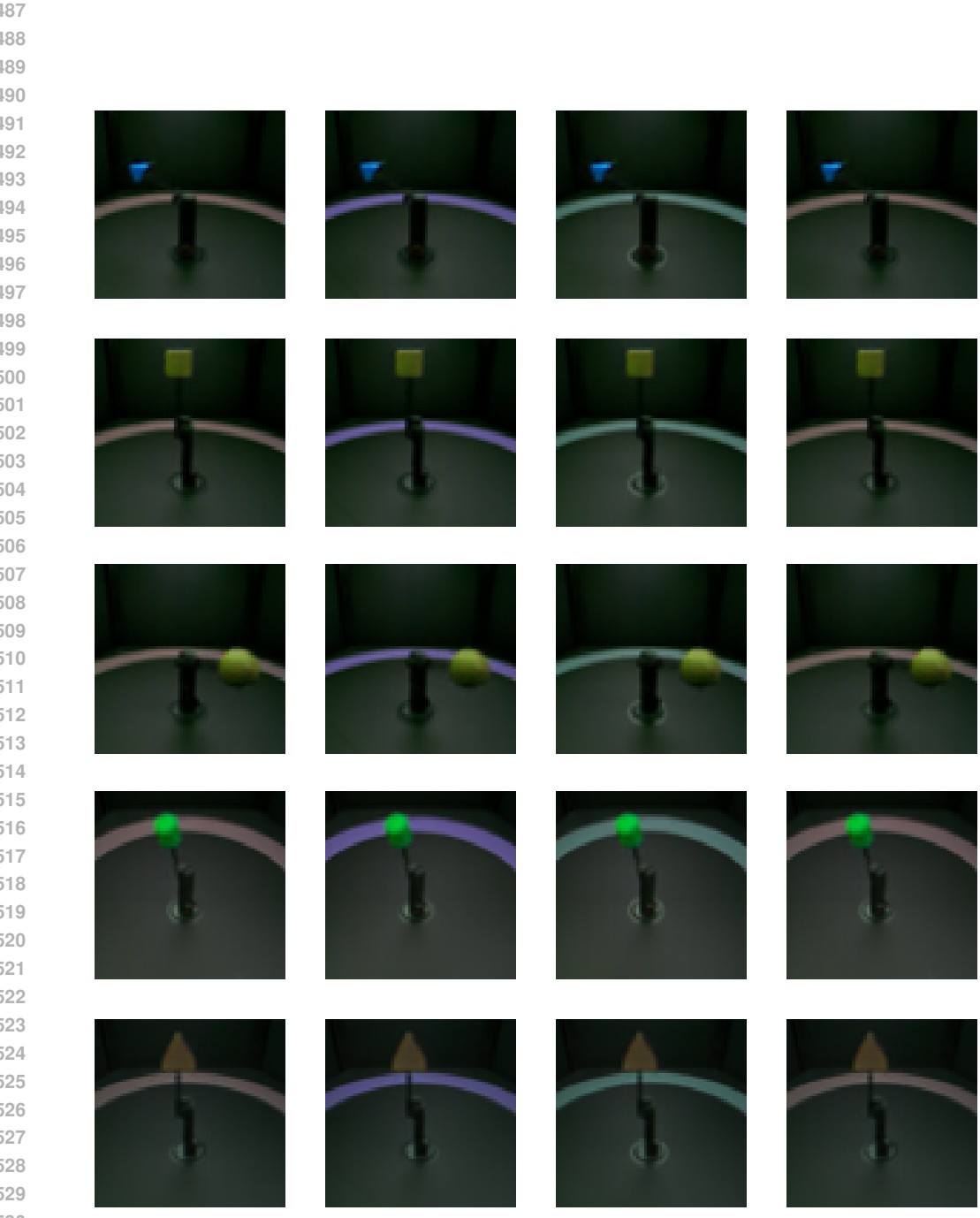

Figure 39: MPI3D: Latent traversal with atomic group actions, background color

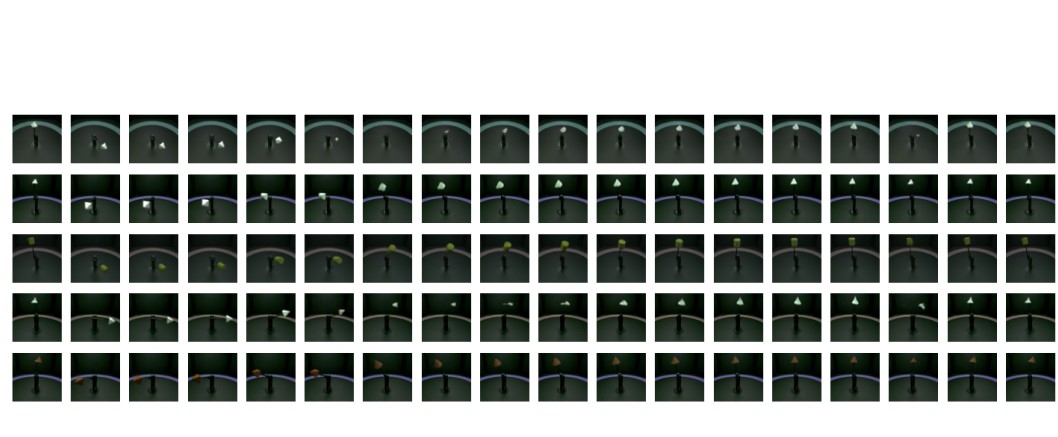

Figure 40: MPI3D: Latent traversal with atomic group actions, horizontal axis

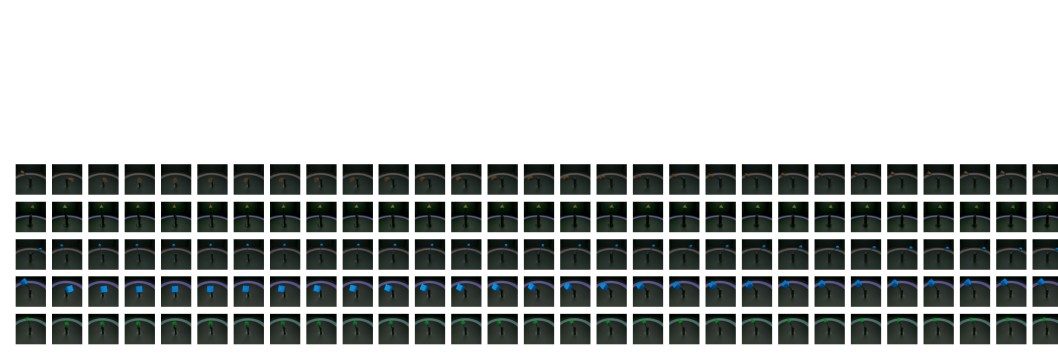

Figure 41: MPI3D: Latent traversal with atomic group actions, vertical axis

