# OpenReview forum: "The Sky Is The Limit When Clustering Is Equated With Disentanglement"
_ICLR.cc/2025/Conference — ICLR 2025 Conference Withdrawn Submission_

### Official Review · Reviewer_cnPY · 2024-10-28

**Soundness:** 1
**Presentation:** 2
**Contribution:** 2
**Rating:** 3
**Confidence:** 3

**Summary:**

This paper introduces a novel method called implicit clustering, which leverages disentangled representations to identify the different factors of variation within a dataset. By focusing on specific factors, the approach implicitly enables the discovery of distinct clusterings without defining a sample similarity function a priori. Additionally, the method enables conditional sampling of synthetic datasets based on desired factors of variation. The authors also demonstrate that their approach establishes a lower bound for clustering.

**Strengths:**

- The ability to cluster without requiring predefined sample similarity is especially valuable for datasets where similarity is difficult to define.
- The experiments offer a comprehensive evaluation of the method's underlying assumptions and demonstrate their effectiveness in practice.

**Weaknesses:**

- The notation throughout the paper is somewhat confusing and overloaded, particularly in Sections 2.1 and 2.3.
- The method relies on the assumption of disentangled representations, which is a strong and often unrealistic requirement in practice.
- All experiments are conducted on synthetic datasets, despite briefly discussing the method's potential in unsupervised settings without access to the ground-truth factors of variation (lines 250-256).
- The experiments lack baselines, and there are no comparisons of the clustering performance or generative capabilities against existing approaches.
- Although the authors mention a more intuitive method for visualizing disentanglement, they do not explain how to interpret the atomic group action plots they present.

**Questions:**

- What exactly do the authors mean by their lower bound? Specifically, how is the operator $<\approx$ defined, and what does it imply when one set is the lower bound of another, as in Equation (3)?
- The overloading of notation makes the manuscript difficult to follow at times. For example, the atomic group action and the generator are denoted by $G$ and $G_\theta$, respectively, and $\mathcal{D}_i’$ has two different definitions in equations (2) and (3).
- The precise definition of the function $\cdot(\cdot,\cdot)$ seems to be missing.
- Despite being mentioned as a key contribution, the term "combinatorial generalization" is never defined.
- The paper is missing the reference to the original VAE paper [1].
- What is the difference between simply clustering in the latent space of a VAE and the approach used in the experiments for Table 2?
- Did the authors use a train-test split for the results in Table 2? Are these results reported on the train or test set?
- Can the authors provide a table showing clustering results across all factors of variation and datasets?

[1] Kingma, Diederik P., and Max Welling. ‘Auto-Encoding Variational Bayes’. 2nd International Conference on Learning Representations, ICLR 2014

---

### Official Review · Reviewer_SHLG · 2024-10-29

**Soundness:** 2
**Presentation:** 2
**Contribution:** 2
**Rating:** 3
**Confidence:** 4

**Summary:**

This paper presents a theoretical and practical exploration of equating disentangled representation learning with clustering by leveraging factors of variation. The authors propose Implicit Neural Clustering, a method that clusters data points by sampling and generating elements implicitly through controlled factors of variation in a disentangled latent space. By doing so, they suggest a lower bound on clustering, derived from the encoder's capacity for disentanglement and the generator’s combinatorial generalization abilities. The approach is validated experimentally on well-known datasets (e.g., dSprites, 3DShapes, MPI3D) and demonstrates how implicit clusters could synthesize realistic variations in data, potentially filling gaps in existing data distributions, enhancing interpretability, and reducing data storage needs.

**Strengths:**

- Innovative Theoretical Insights: The paper introduces a fresh perspective by equating clustering with disentanglement, suggesting that controlled, disentangled representations can generate implicit clusters with practical applications in dataset synthesis.
- Applicability to Real-World Problems: The concept has potential applications in data augmentation, interpretability, and data-efficient machine learning. This approach could impact areas where data acquisition is expensive or limited, such as medical imaging.

**Weaknesses:**

- Complexity of Methodology:
    - Certain sections, such as the detailed sampling procedure and atomic group actions, could be challenging to follow without prior knowledge of disentangled representation theory. A more detailed explanation or visual examples of the atomic group actions and the sampling procedure would aid readers unfamiliar with disentangled representation learning.
- Ambiguity in Practical Applicability:
    - While the theoretical insights are compelling, the practical scalability of Implicit Neural Clustering on complex datasets (e.g., CIFAR-10 or ImageNet) remains unclear. Demonstrating the method’s performance on a higher-dimensional dataset or discussing scalability and computational requirements would strengthen the paper's impact.
- Limited Scope on Combinatorial Generalization:
    - The manuscript mentions the challenge of achieving combinatorial generalization but provides limited experimental analysis on how well Implicit Neural Clustering can handle novel combinations. Additional experiments showcasing how well the approach generalizes to unseen combinations of factors of variation would highlight its effectiveness in real-world scenarios.
- Strong Assumptions:
   - - The assumption of having a disentangled representation is strong. Would you happen to know if it will hold in practice? The authors already write in the abstract that the promises of disentangled representations could not yet be fulfilled on real-world datasets and could further limit the practical applicability to real-world scenarios.

**Questions:**

- Could you explain the plots for visualizing disentanglement in more detail? They are not clear to me.
- while presenting novel measures for disentanglement is interesting, they seem to need knowledge about the ground truth factors of variation in the data. How would you take this measure to real-world datasets where this knowledge is not available? or only partly available?

---

### Official Review · Reviewer_7Hty · 2024-10-31

**Soundness:** 3
**Presentation:** 2
**Contribution:** 2
**Rating:** 5
**Confidence:** 3

**Summary:**

This paper introduces Implicit Neural Clustering, a method that aims at generalization the traditional notion of explicit clustering by taking into account disentanglement. The authors propose generating clusters implicitly by manipulating factors of variation in a disentangled latent space, moving beyond traditional explicit clustering methods. The authors argue that that such sampling procedure would enable generalizing traditional explicit clustering from finite dataset by generating novel elements through combinatoral generalization.

The authors claim that such strategy may enable synthesizing exhaustive datasets from limited data, thus improving interpretability in cluster analysis, enhancing classification tasks in the case of imbalanced data, and reducing data storage needs. The authors performed experiments on synthetic datasets (dSprites, 3DShapes, MPI3D) and demonstrated the ability to form multi-partitioning clusters given a disentangled latent space, without having to assume the number of clusters a priori.

**Strengths:**

The paper introduces Implicit Neural Clustering, offering a fresh perspective that equates clustering with disentangled representation learning, which has the potential to advance the field significantly and provides a theoretical framework that clarifies the relationship between clustering and factors of variation, enhancing the overall understanding of both concepts.

The experiments conducted on synthetic datasets (dSprites, 3DShapes, MPI3D) effectively demonstrate the practical applicability of the proposed method, showcasing its ability to generate meaningful clusters. These experiments also put the emphasis on combinatorial generalization highlights important implications for dataset synthesis, broadening the potential use cases for the approach.

Overall, and for the most part, related works have been properly adressed.

**Weaknesses:**

Overall, the experimental section of the paper appears limited, relying solely on three synthetic datasets. The inclusion of real-world natural or medical imaging experiments would strengthen the findings. Additionally, the implicit clustering sampling procedure is tested only within the Variational Auto-Encoder (VAE) latent space, which may not represent state-of-the-art techniques today. A discussion of this limitation, along with suggestions on how the implicit clustering method could be applied to more recent generative models, would enhance the paper's relevance.

Furthermore, it may be challenging for readers to differentiate the proposed sampling procedure from a simple latent space traversal sampling. A comparison with such a straightforward method could help illustrate the strengths of the proposed approach.

On page 6, the authors assert that their qualitative visualization technique surpasses Hinton Matrices. However, further elaboration on this claim would be beneficial. Including a quantitative measure to demonstrate how their method captures more fine-grained information could strengthen this argument.

Lastly, the technique relies on the assumptions that (1) the encoder generates an identifiable latent space and (2) the latent space is disentangled. These assumptions are not fully resolved in the context of self-supervised methods. Discussing how these assumptions impact implicit clustering could provide valuable insights.

**Questions:**

What do the authors think about the recent paper proving that using artificially generated samples as input to training generative models may lead to degenerate results ? [A]

S. Alemohammad, Self-Consuming Generative Models Go MAD, ICLR 2023, https://arxiv.org/abs/2307.01850

Do you need to make any assumptions about the form of the latent space distributions when fitting a KDE ? How does this impact the sampling procedure ?

---

### Official Review · Reviewer_GpKR · 2024-10-31

**Soundness:** 2
**Presentation:** 3
**Contribution:** 2
**Rating:** 5
**Confidence:** 3

**Summary:**

The paper presents Implicit Neural Clustering, which is a sampling method for generating clusters implicitly through disentangled representations. The proposed method implicitly clusters instances of a dataset by learning to represent and generate the elements of each cluster. Compared to other methods it does not explicitly cluster the elements of a dataset and allows the generation of unobserved combinations of factors of variation.

**Strengths:**

**Originality:**
- Apart from the proposed method, the paper presents a novel angle to understand the relationship between disentangled representation learning and clustering, which is an interesting contribution.

**Quality:**
- The paper provides a wide range of quantitative and qualitative experiments with commonly used synthetic disentanglement benchmark datasets.

**Clarity**
- The problem definition is clear and well-presented, including the disentangled representations, implicit neural clustering and the proposed sampling procedure;
- The paper provides all technical details together with pseudo-code and illustrative figures, which make the method understandable and reproducible.

**Significance**
- The proposed method and theory should be of significance to the subfields of disentangled representation learning and deep clustering.

**Weaknesses:**

**Clarity:**
- The underlying motivation of this work should be improved. For example, it is unclear to me how the notion of clustering in the presented work is improving the generation of novel samples. There exist already methods, e.g., diffusion models, that can generate highly realistic images without the notion of clusters or disentanglement.

Typos and minor issues:
- Incomplete citation: “more informative than Hinton Matrices eastwood2018framework, montero2022lost).”
- The discussion in Section 4, should be restructured using subsections. For example, currently the discussed limitations can be easily missed.

**Quality:**
- No experiments on real world data sets are conducted. The authors provide only experiments for synthetic data sets. This is fine for experiments on disentanglement so that ground truth factors of variation are known, but the image generation and clustering aspects should be investigated on commonly used benchmark data sets, like CIFAR10 for image clustering.
- No runtime experiments or runtime complexity analysis of the proposed method is conducted
- Related work on alternative clustering / non-redundant clustering is missing. These are a family of approaches that also find multiple clusterings in a data set.

**Questions:**

**Motivation**

- The underlying real world motivation for this work is unclear to me. Please clarify what can be gained from your method in terms of clustering and image generation performance. I am especially interested where the benefit of this method lies in comparison to existing image generation approaches, like stable diffusion, or deep clustering approaches.

**Image generation**
- Can you illustrate your methods benefit of "implict neural clustering" for image generation on a non-synthetic data set? I would be especially interested if your "synthesis of novel elements" works for more complicated data sets, like subsets of Imagenet or CIFAR10/100. For example, if you have a factor of variation of "color" and observe a grey CIFAR10 image, can your method generate the missing color?

**Clustering Experiments**

- Please clarify what exactly is clustered in Table 2 and which labels are used to compute the clustering metrics (ACC, NMI)?
- How many clusters are found with KDE in Table 2? Does KDE discover the true number of clusters?
- How does your method compare to other existing (deep) image clustering methods on real world data sets like CIFAR10/100?

---

### Official Review · Reviewer_5JS8 · 2024-11-01

**Soundness:** 2
**Presentation:** 1
**Contribution:** 2
**Rating:** 3
**Confidence:** 3

**Summary:**

This paper proposed a method called implicit neural clustering which uses clustering to learn disentangled representations and generate data with unseen combinations of explanatory factors. The proposed method was evaluated on synthetic image datasets (dSprites, 3D Shapes, MPI3D).

**Strengths:**

- Learning from limited combinations of factors and generating novel combinations useen during training is a very important and challenging problem.
- Using generative models may be a good idea for this problem.

**Weaknesses:**

- The presentation is confusing to me. Many terms and expressions in the introduction were unclear and unexplained (e.g., "_theoretically_ equate (?) clustering and generative models with disentanglement", "_explicit_ clusters are a subset of _implicit_ clusters", "implicit clusters have a lower bound", "_atomic_ group actions", "partition of a latent traversal direction"). It's hard to guess what the author is trying to express.
- I don't think the way the term "lower bound" is used here is what I'm familiar with. A bound (in math) is given by an inequality, while it seems that the "lower bound" here is not used in the usual sense. The symbol $<\approx$, if it's not a typo, was not explained.
- The author proposed an "effective and simple qualitative measure for disentanglement" and claimed that it is more informative than the existing ones. However, evidence for its effectiveness and validity is lacking.
- This paper lacks minimal proofreading (e.g., eastwood2018framework, montero2022lost in l.310-311, the appendix looks unfinished.)
- The code is only available upon acceptance, so we cannot check if the claims match the implementation.

**Questions:**

- I'm not against catchy and memorable titles. However, the meaning of the current title is very unclear. Why "the sky is the limit"? This phrase/metaphor has never been used or explained in the main text. What do you mean by "equating disentanglement with clustering" or "equating clustering with disentanglement"? (Perhaps you mean "equipping"?)
- l.260 "systematically increase the number of assumptions"?

---

### Official Review · Reviewer_V3Du · 2024-11-04

**Soundness:** 2
**Presentation:** 2
**Contribution:** 2
**Rating:** 3
**Confidence:** 3

**Summary:**

The paper introduces. a method to perform implicit clustering via disentangled representation learning, rather than explicitly assigning data points to clusters. The approach is interesting, but the method depends on disentangled representation and it is unclear if it can scale to more complex settings and would support the claims made in the paper.

**Strengths:**

- The concept of equating clustering with disentangled representation learning is a new perspective.
- The paper includes multiple ablation studies of the assumptions, which shows the authors’ effort to perform thorough evaluations.

**Weaknesses:**

- The paper mentions multi-dimensional latent modification or combinatorial generalization for the advantage of the method. However, I am not sure if those are unique to the proposed method. Don’t existing disentanglement learning methods based on VAEs, such as factories, also support these properties?
- The paper claims several advantages of the method, including synthesizing complete datasets from limited data, addressing data distribution gaps, improving interpretability in cluster analysis, enhancing SSL and classification tasks, and reducing data storage space. In the discussion, it is even mentioned that the approach naturally extends to real-world SSL settings. However, this seems far-fetched. All tested datasets and tasks in the paper are small toy datasets.
- There have been discussions that unsupervised learning of disentangled representations for real-world data is not possible (e.g., https://arxiv.org/abs/1811.12359) It is unclear whether this is not the case for this proposed method.
- The readability of the paper, especially in Section 2.1,  is low. I believe the readability of the paper, in general, could be improved by presenting the high-level idea and explaining the meaning of theoretical results first rather than just placing long theoretical discussions back-to-back.

**Questions:**

- The paper mentions multi-dimensional latent modification or combinatorial generalization for the advantage of the method. However, I am not sure if those are unique to the proposed method. Don’t existing disentanglement learning methods based on VAEs, such as factories, also support these properties?

---

### Note · Authors · 2024-11-14

I have read and agree with the venue's withdrawal policy on behalf of myself and my co-authors.